Technical Report

# LIANA+ provides an all-in-one framework for cell–cell communication inference

Daniel Dimitrov[1], Philipp Sven Lars Schäfer[1], Elias Farr[1],
Pablo Rodriguez-Mier [1], Sebastian Lobentanzer[1], Pau Badia-i-Mompel [1,2],
Aurelien Dugourd[1], Jovan Tanevski [1], Ricardo Omar Ramirez Flores [1] &
Julio Saez-Rodriguez[1,3] ✉

The growing availability of single-cell and spatially resolved transcriptomics has led to the development of many approaches to infer cell–cell communication, each capturing only a partial view of the complex landscape of intercellular signalling. Here we present LIANA+, a scalable framework built around a rich knowledge base to decode coordinated inter- and intracellular signalling events from single- and multi-condition datasets in both single-cell and spatially resolved data. By extending and unifying established methodologies, LIANA+ provides a comprehensive set of synergistic components to study cell–cell communication via diverse molecular mediators, including those measured in multi-omics data. LIANA+ is accessible at https://github.com/saezlab/liana-py with extensive vignettes (https://liana-py.readthedocs.io/) and provides an all-in-one solution to intercellular communication inference.

Cell–cell communication (CCC) inference has recently emerged as a major component of the analysis of single-cell and spatially resolved transcriptomics data, with over 100 tools contributing valuable developments[1,2]. All single-cell methods are based on multiple assumptions, including that gene co-expression across dissociated cells, or groups of cells, reflects CCC within tissues[3]. Similarly, CCC methods that use spatial information quantify co-localizations at different scales, some summarizing interactions globally, across slides as a whole[4–6], and others locally at the individual cell or location[7–9].

Most methods have focused on protein-mediated interactions, predominantly inferred from transcriptomics data[2,3], and only a few from multi-omics data[10]. As a consequence, other modes of intercellular signalling have been typically ignored[3] except for limited metabolite-mediated CCC predictions from transcriptomics data[11–14]. Emerging multi-omics technologies[15] are anticipated to provide a broader picture of molecular mediators and in turn prompt the development of new CCC tools.

While early methods analysed CCC from single-condition datasets, increasing sample numbers and experimental design complexity have prompted various strategies to extract differential CCC insights. These strategies include methods that (1) consider each interaction independently[8,16,17], (2) make use of dimensionality reduction to perform pairwise comparisons between conditions[18,19] or (3) model all variables, samples and cell types simultaneously[20]. Approaches 2 and 3 can be thought of as modelling orchestrated CCC events, here referred to as 'intercellular programmes'.

CCC methods typically rely on pre-existing knowledge[2,3], with extensive efforts dedicated to curating and extending protein-mediated[19,21,22] and, to a lesser extent, metabolite-mediated knowledge[11,13,14,23]. In some resources, the interactions are associated with pathways[19] or transcriptional regulators[24,25], leading to multiple heterogeneous databases and potential inconsistencies caused solely by the choice of resource[3].

Finally, all these developments use various infrastructures, with each CCC tool being typically designed for a specific task or data type.

Here, we introduce LIANA+, an all-in-one framework that enables CCC inference beyond a single task or data type. To illustrate the distinguishing features of LIANA+, we applied it to a recent spatially resolved metabolome–transcriptome dataset of a murine Parkinson's disease model[26]. In this case study, we identified dopamine-mediated

[1]Faculty of Medicine and Heidelberg University Hospital, Institute for Computational Biomedicine, Heidelberg University, Heidelberg, Germany. [2]GSK, Cellzome, Heidelberg, Germany. [3]European Bioinformatics Institute, European Molecular Biology Laboratory, Hinxton, UK. ✉e-mail: saezlab@ebi.ac.uk

CCC events and the brain subregions where they take place. Moreover, we jointly analysed single-nucleus and spatially resolved human heart data with a complex cross-conditional experimental design[27]. In this analysis, we hypothesized intercellular and intracellular signalling mechanisms driving fibrosis in ischaemic heart regions.

## Results

### LIANA+ as an all-in-one solution to model CCC

LIANA+ is a scalable framework (Extended Data Fig. 1, Supplementary Table 1 and Supplementary Note 1) that integrates methods to infer CCC from dissociated data (Fig. 1a) and methods to study global and local relationships from spatially resolved data (Fig. 1b), expanding them to multi-omics technologies. When working with cross-conditional data-sets, these methods are supplemented by different strategies to extract deregulated CCC events via hypothesis-free and hypothesis-driven approaches (Fig. 1c). LIANA+ uses standardized scverse[28] input and output formats (Fig. 2a), enabling interoperability with external packages and the straightforward extensions of CCC approaches. Moreover, we propose a flexible causal subnetwork search to link CCC events with intracellular signalling (Fig. 1d). Each of these components leverages a comprehensive prior knowledge base that encompasses metabolite- and protein-mediated intercellular and intracellular signalling (Figs. 1e and 2b–e and Supplementary Note 2). Taken together, LIANA+ synthesises heterogeneous methods and resources[1,2], providing an all-in-one framework to study intercellular communication (Fig. 1 and Supplementary Table 2).

### LIANA+ enables modelling CCC across spatial modalities

A particular challenge of emerging spatial technologies is that, on a single tissue section, they can combine different technologies[15], and hence the observations from each technology may correspond to distinct spatial locations that need to be aligned[26,29]. LIANA+ handles spatial multi-omics datasets from diverse technologies, including such with distinct observations across modalities (Methods).

Expanding on previous work[4], LIANA+ uses a multi-view modelling approach to learn spatial relationships across distinct types of features, spatial contexts and technologies (represented as views; Fig. 1b). This approach enables the joint modelling of combinations of complex tissue structures (for example, cell neighbourhoods) and functions (for example, signalling pathways). For example, relationships between ligands and pathways[4] or cell types and pathways[27,30]. Yet it models relationships globally and hence does not provide information about the tissue locations within which these interactions occur. To complement it, we implemented eight local metrics (Fig. 1b and Supplementary Note 3). These have previously been used to identify co-expression patterns between genes across spatial[31–33] and pseudotime[32] contexts, and have been recently applied to ligand–receptor interactions[8,9,34]. We illustrate the joint application of multi-view modelling and local metrics to study metabolite-mediated interactions from multi-omics data using a recent murine Parkinson's disease model dataset[26]. This spatially resolved dataset provides metabolome and transcriptome measurements, respectively generated using matrix-assisted laser desorption/ionization mass spectrometry imaging and 10X Visium technologies[26] (Fig. 3a). Briefly, three mice were subjected to unilateral 6-hydroxydopamine-induced lesions in one hemisphere while the other remained intact[26] (Fig. 3b). These 6-hydroxydopamine-induced lesions selectively destroy substantia nigra-originated dopaminergic neurons, impairing dopamine-mediated regulatory mechanisms in the striatum—an area of the brain crucial for movement coordination (Fig. 3b).

### LIANA+ jointly models global associations across modalities.

Using multi-view modelling, we inferred spatial relationships between metabolites, their corresponding brain-specific metabolite receptors and cell types across different spatial contexts (views) (Fig. 3c and Methods). Specifically, we trained a model that predicts metabolite intensities

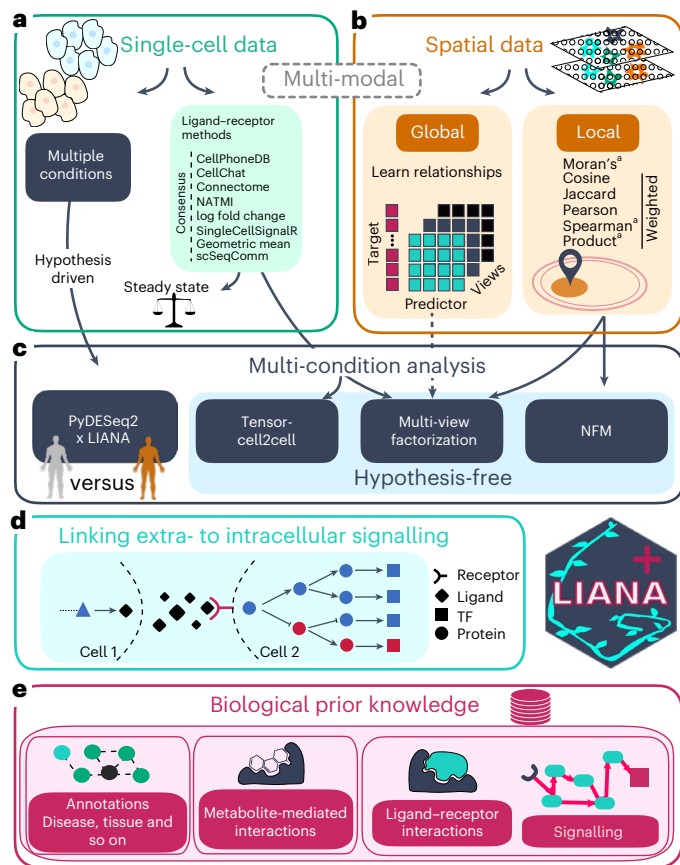

**Fig. 1 | LIANA+ framework overview. a**, LIANA+ re-implements and adapts eight ligand–receptor methods to infer interactions from single-cell data, along with a flexible consensus that can integrate any combination of these methods. **b**, LIANA+ implements multi-view learning as well as eight local metrics to respectively capture global and local interactions from spatially resolved omics data. **c**, LIANA+ includes diverse strategies to identify deregulated CCC events across conditions: (1) differential CCC analysis with PyDESeq2 (refs. 47,48) for hypothesis-driven exploration and (2) unsupervised (hypothesis-free) approaches including standard NMF or higher-order factorizations via Tensor-cell2cell[20] and multi-view factor analysis[43]. **d**, LIANA+ connects intercellular interactions to intracellular signalling pathways using sign-coherent network optimization. **e**, LIANA+ is built on a rich knowledge base—OmniPath[22] and BioCypher[59]—which comprise ligand–receptor interactions and annotations, including such mediated by metabolites[68], as well as intracellular knowledge, such as signalling pathways and TFs. Finally, all components (**a**–**e**) of LIANA+ are applicable to both dissociated single-cell and spatially resolved multi-omics data. ªFor spatially weighted Spearman correlation, along with the standard metric, we implemented its masked version from scHOT[32]; for Moran's $R$ we adapted both the global and local versions from SpatialDM[8] and for the spatially weighted product, we also included a max-normalized version.

using spatially adjacent receptor expression and cell-type proportions, deconvoluted using Tangram[35] with a murine brain single-cell atlas[36] as a reference. We then quantified the strength of association (importance) between the metabolite intensities and their predictors. We further calculated the individual contribution of each predictor view as well as the performance of the multi-view model in explaining the variance of the metabolites' intensities (Methods). For each slide, we carried out this modelling process in the lesioned and intact hemispheres independently (Methods).

Several metabolite peaks were explained relatively well by the joint model ($R^2 > 0.5$), including dopamine and its derivative 3-methoxytyramine, among other potentially deregulated, but unannotated, metabolite peaks (Extended Data Fig. 3a). Focusing on dopamine,

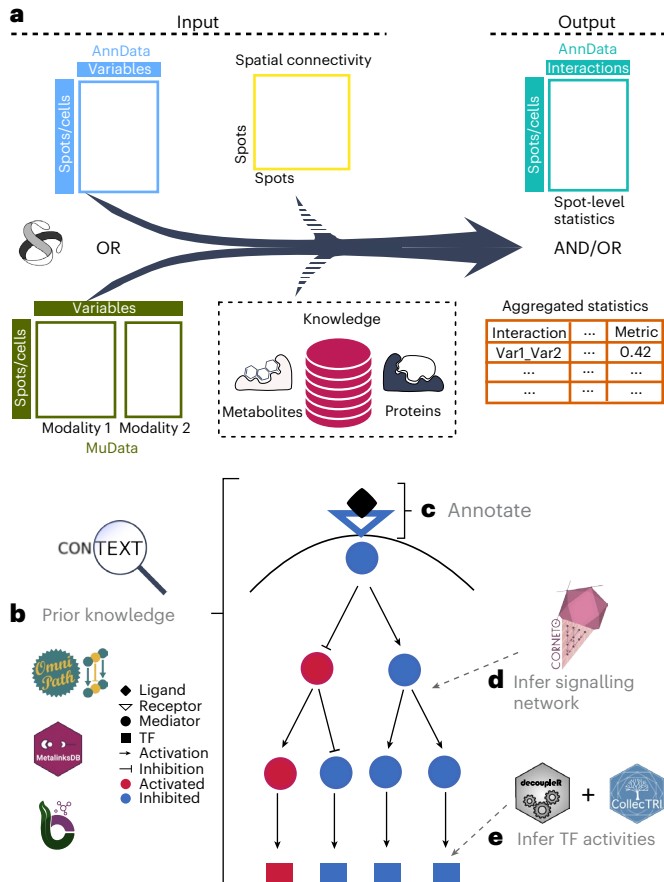

**Fig. 2 | LIANA+ uses standardized inputs and outputs to streamline the inference intercellular and intracellular signalling. a**, LIANA+ accepts unimodal (AnnData) or multi-modal (MuData) data objects as inputs with (optional) prior knowledge and/or spatial information. These are then transformed into data frames with aggregated interaction results or unimodal objects with statistics at the individual spot or cell level. To enable the inference of CCC across modalities, the methods implemented in LIANA+ accept MuData objects[77] as input. These provide essential functionalities to load and store multi-modal data[77], and can be thought of as an extension of AnnData objects[78], which are the default input of LIANA+ when working with unimodal single-cell or spatial data. **b,c**, LIANA+ makes use of existing prior knowledge frameworks (**b**)[22,59,68] to annotate interactions according to, for example, pathways, disease or location (**c**). **d**, Similarly, it uses this prior knowledge to infer putative causal (sign-coherent) signalling networks[49,79], emanating from ligand–receptor interactions down to active TFs. **e**, The TF activities can be estimated by making use of generalistic regulon prior knowledge[80] and standard enrichment analyses[81].

we saw a large difference in explained variance between the intact ($R^2 = 0.535$) and lesioned hemispheres ($R^2 \approx 0$; Extended Data Fig. 3a), which is expected due to the absence of dopamine in the striatum of the lesioned hemisphere[26] (Fig. 3e and Extended Data Fig. 4). Looking further into the intact hemisphere, we saw that the cell-type proportions were a better predictor of metabolite intensities than the receptors (Extended Data Fig. 3b,c and Methods), implying that dopamine signalling in this region was more closely associated with the abundance of specific cell types than the expression of brain-specific metabolite receptors. Similarly, we noted that cell types were typically better predictors than receptors for the remainder of the well-explained metabolite peaks (Extended Data Fig. 3b). Focusing on the intact hemispheres, we found that the three best predictors of dopamine (median $t$-value of >3) were dorsal medium-sized spiny neurons (MSNs) 1 and 2 (ref. 36) and *Drd2* dopamine receptor gene expression (Fig. 3d). This reflects anticipated associations with dopamine, as D2R (encoded

by *Drd2*) is a canonical receptor of dopamine. Similarly, GABAergic MSNs 1/2 are key receivers of dopamine signalling and were classified as types D1 or D2 according to the expression of dopamine receptor genes (*Drd1* or *Drd2*, respectively; Fig. 3f and Extended Data Fig. 4d) they express[36]. Moreover, dopamine's relationship with its top three predictors differed notably between intact and lesioned hemispheres (Extended Data Fig. 3d–f).

This association of dopamine with the MSN1 cell type corroborated the findings of the original publication[26], while our analysis highlighted the interactions of dopamine with MSN2 and its canonical D2 receptor, which were not previously reported there[36].

**LIANA+ infers local interactions at individual locations.** The approach described above models global spatial relationships—that is, it considers all spots to compute a single value per interaction across the slide. As such, it provides a single statistic for the importance of each interaction in a slide but does not provide information about the region or locations where the interactions occur. To complement global relationships identified with LIANA+, we implemented eight metrics to pinpoint local interactions at the individual spot or cell location. Briefly, LIANA+ includes (1) four spatially weighted variants of commonly used similarity metrics (cosine similarity, Pearson and Spearman correlation and Jaccard index), (2) a masked version of Spearman correlation[32], (3) simple spatially weighted products and (4) a bivariate extension[8,33,37] of the univariate spatial clustering measure—Moran's I (Methods). We evaluated the performance of these metrics in two separate tasks (Extended Data Fig. 2 and Supplementary Note 4), and chose spatially weighted cosine as the default local metric in LIANA+ due to its interpretability and consistent performance. Along with the local metrics, we further provide local permutation $P$ values and categories, the latter reflecting whether an interaction between the two variables is positive, negative or neither (Methods).

Using spatially weighted cosine similarity, we focused on identifying the specific locations at which putative interactions with dopamine occurred. We saw that, as anticipated, the putative interactions of dopamine and D2R, highlighted above by the global multi-view learning approach, occurred only within the intact striatum regions (Fig. 3g), which was further corroborated by low $P$ values (Fig. 3h) and a positive association (category) between the two variables (Fig. 3i). With similar results for MSN1 and MSN2 cell types (Extended Data Fig. 4e,f). Our analysis additionally hinted at an anticipated asymmetry following uni-hemispheric lesion[38]: while the interaction between D2R and dopamine was present in the intact hemisphere (Fig. 3i), *Drd2* was also expressed in the dopamine-depleted, lesioned hemisphere (Fig. 3f).

In conclusion, using LIANA+ we captured perturbation-driven changes in dopamine's distribution and its associations with its canonical D2R receptor and MSN cell types 1 and 2. We also pinpointed the specific regions where these interactions take place, recapitulating and extending perturbed dopamine-signalling hypotheses[26] and illustrating how LIANA+ enables CCC analyses from spatial multi-omics data.

**Ligand–receptor inference weakly reflects co-localization**

Identifying co-localized genes from spatially resolved data alone can help us to pinpoint relevant interactions driving disease. However, it remains limited by a common coverage–resolution trade-off in most spatial technologies[39,40]: they either quantify a limited fraction of molecules or they capture multiple cells within spots, relying on deconvolution to quantify cell-type frequencies within them[39,40]. Leveraging Slide-tags, a recent technology capable of measuring full transcriptome single-nucleus data while also preserving spatial information[41], we evaluated spatially uniformed CCC methods using cell type and gene expression co-localization as an indirect ground truth[3]—assuming that co-localization is a proxy of communication (Methods). We found weak associations between ligand–receptor interactions predicted in a spatially agnostic manner and the co-localization of their ligands,

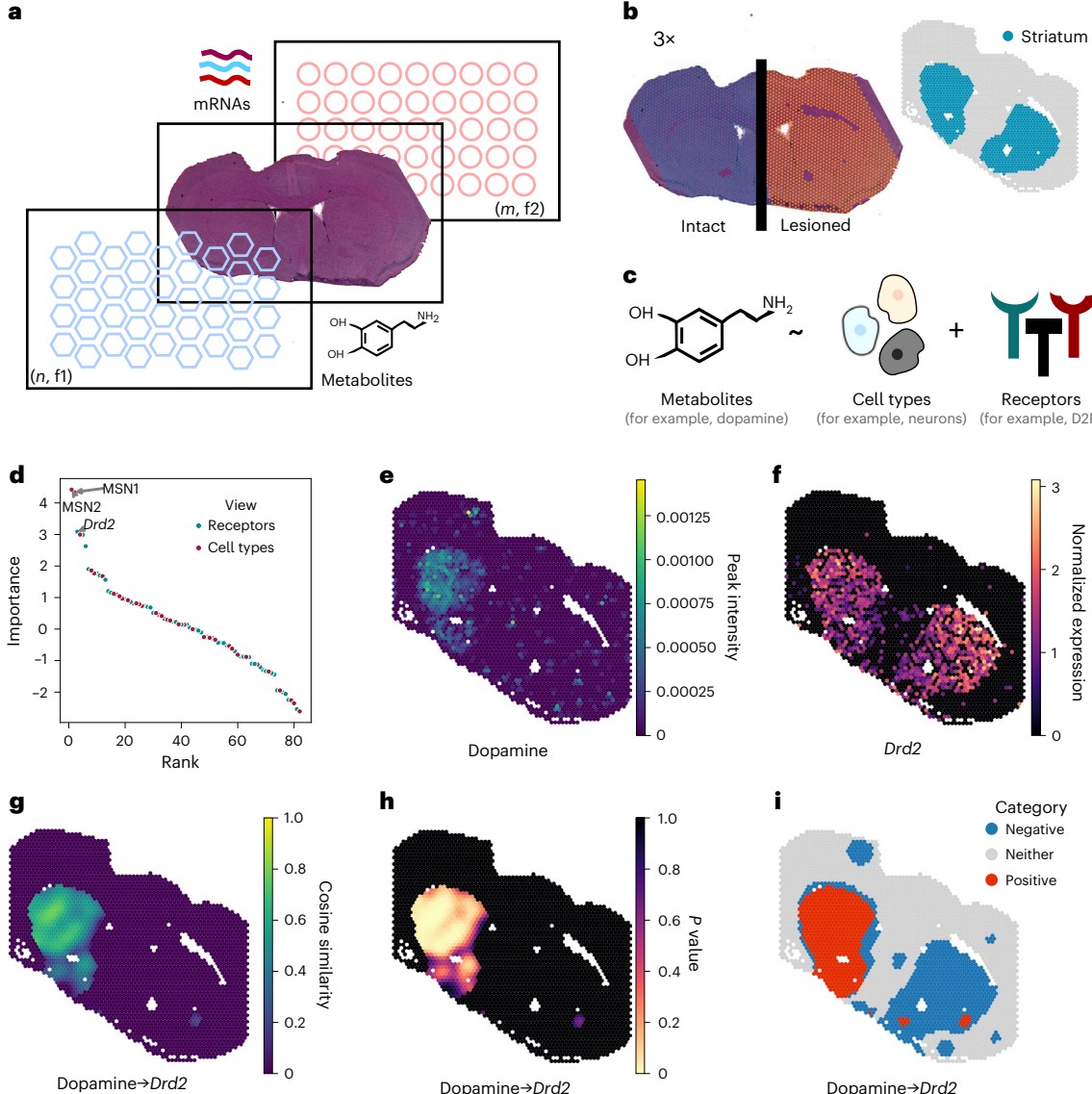

**Fig. 3 | LIANA+ models intercellular communication from spatial multi-omics data. a**, Spatially resolved transcriptomics (10X Visium) and metabolomics (matrix-assisted laser desorption/ionization mass spectrometry imaging) yield two matrices with different sets of features for each modality (f1 or f2), and observations (n or m) which correspond to different locations captured on the same tissue section[26]. **b**, Parkinson's disease mouse model annotated for striatum in intact and lesioned hemispheres, with three replicates (3×). **c**, Multi-view modelling integrates metabolite peak intensities, brain-specific receptor expression and cell-type proportions to identify their spatial relationships. This approach enables the estimation of joint performance and individual contributions of receptor expression and cell-type proportions in predicting metabolite peak intensities. Cell-type proportions were deconvoluted[35] using a murine single-cell atlas[36] as a reference. **d**, Dopamine predictors ranked according to their median importance (y axis; ordinary least squares t-values), with names shown for the top three predictors: Drd2 and MSNs 1 and 2. **e**, Normalized dopamine peak intensities. **f**, log1p Drd2 receptor gene expression. **g–i**, Local interactions between dopamine and its canonical D2R receptor, encoded by Drd2, as measured by spatially weighted cosine similarity (**g**), its corresponding uncorrected permutation P values (**h**) and interaction categories (**i**). The images showcase slide B1 from experiment V11L12-109 (ref. 36). Source numerical data are available in source data.

receptors and cell types (Fig. 4 and Supplementary Note 5). Therefore, in the next section, we combined dissociated and spatial data to contextualize CCC inference to proteins and cell types known to co-localize.

## LIANA+ detects deregulated inter- and intracellular events

With growing sample sizes and increasingly complex experimental setups, versatile methods are required to analyse CCC across conditions. Standard dimensionality reduction approaches, such as non-negative matrix factorization (NMF) or principal component analysis, have been previously used to investigate cross-talk between cell types[18,19,42]. However, such approaches are limited to capturing the variance across data with only two dimensions at a time, for example, observations

and interactions. To address this, LIANA+ couples nine ligand–receptor methods (Fig. 1a and Supplementary Table 3) with higher-order factorizations[20,43] for the scalable inference of cross-conditional CCC from dissociated single-cell data (Fig. 1c). The combination of ligand–receptor inference with such factorization methods allows all interactions, cell types and samples to be considered simultaneously[20], thus enabling the identification of coordinated CCC events, or intercellular programmes, across conditions. Moreover, such approaches also highlight the specific interactions and cell types that contribute to these differences, thus offering a complete overview of the elements driving the variation across samples. Along with Tensor-cell2cell[20,44], which uses tensor-based factorization to decompose CCC across samples[20],

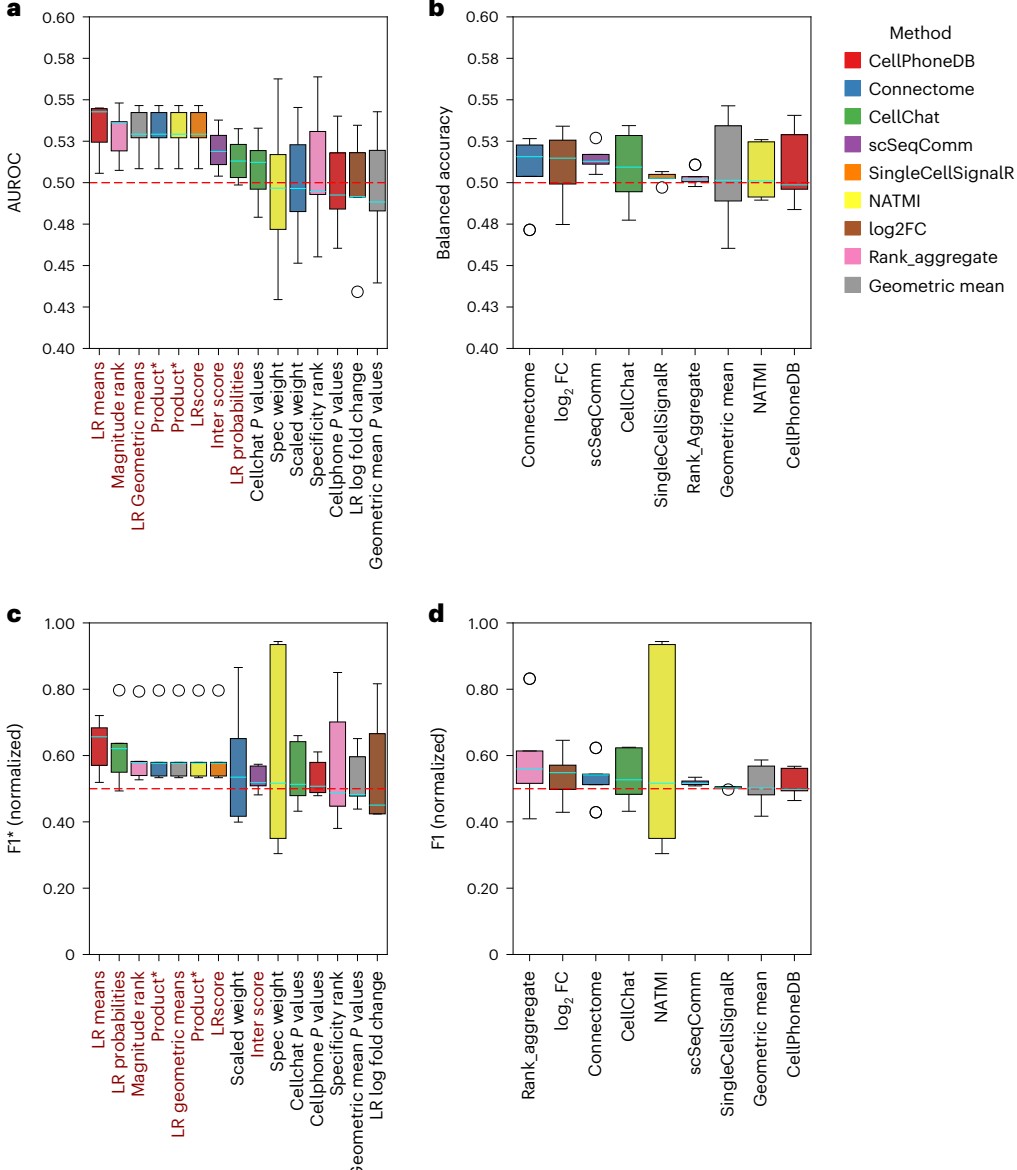

**Fig. 4 | Comparison of ligand–receptor inference methods using the spatial co-localization of cell types and ligand–receptors as assumed truth.** Each method, as re-implemented in LIANA+, and its individual ligand-receptor (LR) scoring functions are represented by a different box colour. Scoring functions that reflect the magnitude strength of an interaction are shown in dark red, while those that capture cell-type specificity in black. **a**, Quantification of the performance of each method's scoring functions using the AUROC. **b**, The balanced accuracy for each of the methods filtered according to their suggested false positive filtering thresholds. **c,d**, The normalized F1 scores following filtering for each score (*) (**c**) and method (**d**). For each metric, a score of 0.5,

denoted by the dashed red line, indicates random performance. Note that while we show 0 to 1 for normalized F1, unlike balanced accuracy and AUROC, it is bound between negative and positive infinity (Methods). Both NATMI and Connectome use expression products (Product*) as a measure of magnitude strength. The boxes were ordered according to the median performance across datasets (central line in cyan within each box; $n$ = 5 datasets). The box hinges represent the first and third quartiles, and the whiskers extend up to 1.5 times the interquartile range above and below the box hinges. Outliers are depicted as individual hollow points beyond the whiskers. Source numerical data are available in source data.

---

we implemented in LIANA+ an unsupervised approach that leverages the flexibility of Bayesian multi-view factor analysis[43] (Methods and Supplementary Note 6). This approach has as a distinguishing feature the ability to obtain interaction importances per cell-type pair (view) rather than their global importance across all cell types[20]. Using five public cross-conditional atlases (Supplementary Table 4), we show that, regardless of the ligand–receptor method, both Tensor-cell2cell and multi-view factorization capture intercellular programmes that separate samples according to different conditions (Supplementary Note 6 and Extended Data Fig. 5). To complement the aforementioned unsupervised analyses, LIANA+ uses state-of-the-art[45,46] differential

expression analysis[47,48] to enable the targeted (hypothesis-driven) exploration of CCC across conditions (Methods). Briefly, LIANA+ combines ligand–receptor prior knowledge with differential statistics, calculated at the pseudobulk level, to identify interactions deregulated across cell-type pairs[16] (Methods). Furthermore, since intercellular CCC events and intracellular signalling are intertwined, LIANA+ provides a distinctive strategy to predict signalling pathways up- or downstream of CCC events, combining its comprehensive knowledge base with a causal subnetwork search (Fig. 1d and Supplementary Note 7). Specifically, we solve a network optimization problem[49] that identifies a subnetwork with a signal flow consistent with the changes in

CCC and transcription factors (TFs) (Methods). A distinct feature of our causal subnetwork search is the consideration of sign coherence, thus providing solutions in line with the activator or inhibitory effects of interactions reported in the literature.

To demonstrate the synergistic nature of the components in LIANA+, we analysed a cross-conditional dataset integrating single-nucleus and spatial transcriptomics data. This dataset comprised 29 single-nucleus and 28 10X Visium spatial samples from human myogenic, fibrotic and ischaemic heart regions following myocardial infarction[27] (Fig. 5a). Using these data, we had previously shown the role of myofibroblasts and macrophages in fibrosis, characterized by the synthesis of extracellular matrix proteins for scar tissue formation[27]. Here, we further elucidate the intercellular and corresponding intracellular signalling mechanisms facilitating cardiac tissue repair and remodelling.

**LIANA+ infers intercellular programmes from spatial data.** To identify intercellular programmes shared across the conditions, we first inferred local ligand–receptor interactions using spatially weighted cosine similarity for each of the 28 spatial transcriptomics slides. Then, we concatenated the resulting interactions across all slides and applied NMF, obtaining five intercellular programmes (factors) across all slides and conditions (Methods). Each factor resulted in scores per observation (spot), along with associated interaction loadings (Fig. 5b). We saw that factors 1, 2 and 5 were enriched in ischaemic samples, and factor 4 was enriched in fibrotic and downregulated in (healthy) myogenic samples. In contrast, factor 3 was positively associated with the myogenic condition (Fig. 5c). To elucidate the functional processes associated with the identified factors, we assessed whether ligand–receptor interactions with high loadings were enriched in a set of canonical pathways[50] (Methods and Fig. 5d). We saw that interactions from ischaemia-associated factors 1 and 2 were enriched in hypoxic, epidermal growth factor receptor (EGFR), mitogen-activated protein kinase (MAPK) and JAK-STAT pathways, reflecting anticipated inflammatory patterns in ischaemic regions[51]. Moreover, we noted that interactions in both the ischaemia-associated factor 4 and the fibrosis-associated factor 5 were enriched in transforming growth factor (TGF)-β signalling—a well-known driver of fibrosis[52]. Both factors 4 and 5 were negatively associated with pro-inflammatory pathways, such as tumour-necrosis factor-α (TNF-α) and NFK-β, suggesting the semi-orthogonal (independent) positioning of the interactions in the latent space concerning factors 1 and 2. Similar downregulation of pro-inflammatory pathways was noted in the myogenic-associated factor 3, probably related to the reduced presence of immune cells within the functioning cardiac muscle tissue. As expected, ischaemia-associated factor 1 localized to ischaemic regions (Fig. 5e). Among the top 30 interactions were several integrins (*ITGB1*, *ITGAV*, *ITGA5* and *ITGA7*) interacting with pro-fibrotic (*FN1* and *SPP1*) gene markers[27,53] and matrix glycoproteins (*TNC* and *THBS1*) (Extended Data Fig. 6a). These interactions of integrin complexes with TNC are in agreement with the high expression levels of *TNC* found in other studies that also focus on the early stages of myocardial infarction[54]. *FN1* has been reported as a marker of myofibroblasts[27], while *SPP1* (ref. 27), *FN1* (ref. 55) or both[53] as markers of pro-fibrotic macrophages. Moreover, *THBS1* and *ITGB1* have recently been implicated in a self-amplifying, immune-cell recruitment loop, including FN1+ *THBS1*-expressing macrophages[55]. Overall, the identified intercellular programmes aligned well with the existing literature on pro-fibrotic response upon myocardial infarction, including potential interactions of FN1 and SPP1 with ITGB1-containing complexes within ischaemic regions (Fig. 5f,g).

**LIANA+ infers intercellular programmes from dissociated data.** To investigate fibrosis-associated intercellular programmes further, we performed an unsupervised CCC analysis on the complementary dissociated single-nucleus data, enabling the exploration of specific cell types involved in intercellular signalling. We additionally incorporated co-localization information to mitigate the anticipated high false positive rates (Supplementary Note 5). In particular, we considered interactions only if they were identified as condition relevant by the NMF analysis, and if they occurred between cell-type pairs observed to co-localize (Extended Data Fig. 6b and Methods). Then, we inferred ligand–receptor interactions with LIANA+ for each of the 29 single-nucleus samples, followed by multi-view factorization[4] on the obtained ligand–receptor interactions across all cell-type pairs and all samples (Fig. 5h and Methods). This unsupervised approach identified a factor (factor 1), whose sample loadings were significantly different across the conditions ($P = 9.9 \times 10^{-12}$; Fig. 5i). Moreover, this approach enabled us to quantify how well the variance within each cell-type pair is captured by each of the factors (Methods). We saw that within factor 1, interactions with fibroblasts (FB) were prominent sources of CCC, including the FB-to-myeloid signalling axis ($R^2 = 25.1\%$) (Fig. 5j), which was also consistent with a strong spatial association between the two cell types (Extended Data Fig. 6b). Moreover, among the interactions with the highest loadings, which were predominantly emitted by FB, we observed interactions involving collagens, integrins and lamins, and FN1 (Extended Data Fig. 6c). In summary, this unsupervised and synergistic analysis highlighted the importance of FB and myeloid cells (MY) as potential drivers of the disease trajectory following early myocardial infarction in both single-cell and spatial data. Thus, our findings pointed to a necessity for an in-depth exploration of FB and MY interactions, concurrent with their known role in the fibrotic process in cardiac tissues upon myocardial infarction[27].

**LIANA+ identifies deregulated intracellular signalling.** To investigate the deregulation of specific ligand–receptor interactions in ischaemia along with downstream signalling that originates from these interactions, we performed differential expression analysis, using both myogenic and fibrotic samples as reference (Fig. 5k and Methods).

We saw deregulation of FB-to-MY signalling and the associated ligand–receptor interactions, in agreement with the preceding unsupervised analyses. Examining the individual ligand and receptor genes, we saw that the expression of *THBS1* and *TNC* was significantly deregulated (false discovery rate (FDR) <0.05) in FB and MY, respectively, while *SPP1*, *THBS1* and *FN1* were deregulated in both (Fig. 5l). Additionally, genes involved in ITGB1-containing integrin complexes, interacting with FN1 and SPP1, were upregulated in MY (Fig. 5l). Building on these findings, we delved into the potential intracellular signalling pathways within MY triggered by the FN1/SPP1 and ITGA5 and ITGB1 integrin complex interactions (Fig. 5k). This focus was motivated by the reported role of these proteins in fibrosis[27,53,55] and the co-deregulation of their genes in myeloid and fibrotic cell types (Fig. 5l) as well as the FB-to-MY signalling axis highlighted by both the multi-view factorization latent space (Fig. 5i) and spatial analysis (Extended Data Fig. 6b). Using a signed and directed prior knowledge network[22] (Methods), we linked the predicted interaction between FN1/SPP1 and ITGA5 and ITGB1 with deregulated downstream TFs (Fig. 5k and Methods). This hinted at a putative signalling network involving kinases MAPK1 and MAPK14, and TF co-regulators ATM, EP300 and YAP (Fig. 5m,n). Specifically, these regulatory proteins were predicted to upregulate SMAD1/3 TFs, key members of canonical TGF-β superfamily signalling[56]. We also noted the downregulation of FOXO3, which is potentially mediated by SMAD3-dependent TGF-β1 signalling in cardiac myofibroblasts[57]. Finally, we cross-checked the predictions from our analysis in public heart failure atlases (Supplementary Table 5) and saw that using LIANA+ to integrate single-cell and spatial data improved the reliability of predictions (Supplementary Note 8 and Extended Data Fig. 7).

In conclusion, our analysis generated mechanistic hypotheses about the intercellular and intracellular events linked to the establishment of the myofibroblast phenotype and recruitment of pro-fibrotic SPP1+ macrophages in myocardial infarction[27].

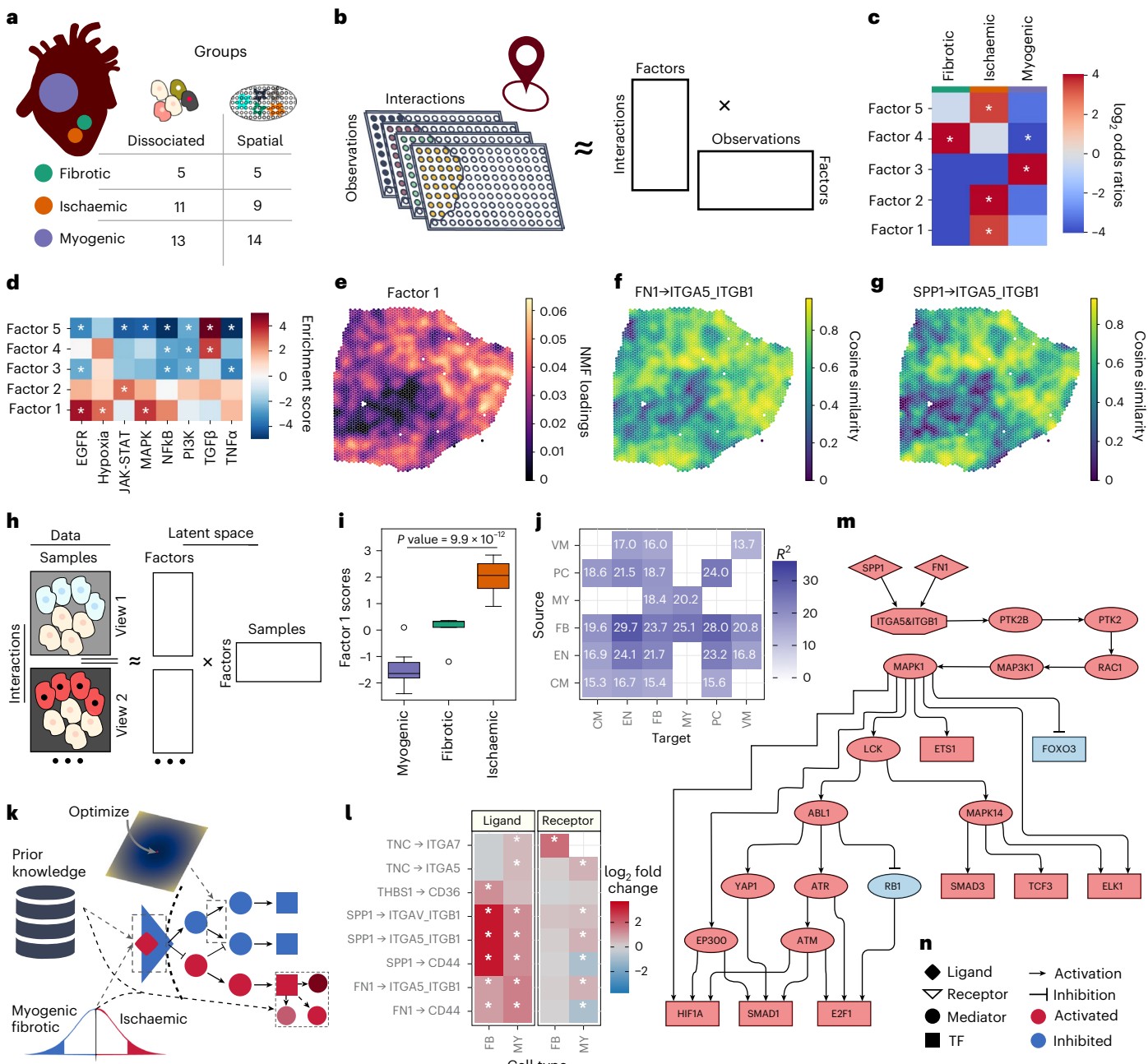

**Fig. 5 | LIANA+ models intercellular communication from dissociated and spatially resolved transcriptomics data. a**, Sampling sites from human myocardial infarction patients. **b**, Overview of NMF applied to local interactions inferred for each location/observation across all slides. **c**, log2-transformed odds ratios and Fisher's exact test *P* values representing the enrichment or depletion of fibrotic, ischaemic and myogenic labels in each of the factors inferred by NMF. **d**, Pathway enrichment[50] of NMF ligand–receptor loadings. Asterisks indicate FDR-corrected *P* values <0.05, with the colour map distinguishing between positive (red) and negative (blue) enrichment. **e**, NMF factor 1 scores per observation in an ischaemic tissue section. **f,g**, Spatially weighted cosine similarity of FN1 (**f**) and SPP1 (**g**) with the ITGA5 and ITGB1 complex, respectively, in a selected (ACH0014) sample. **h**, The procedure to decompose ligand–receptor interactions inferred between cell types from dissociated single-nucleus data samples into factors and corresponding feature sets. **i**, Multi-view factor scores following ligand–receptor score decomposition with one-way analysis of variance *P* value across ischaemic, fibrotic and myogenic samples (*n* = 29). The central line within each box marks the median, with the box hinges representing the first and third quartiles. The whiskers extend up to 1.5 times the interquartile

range above and below the box hinges. Outliers are depicted as individual hollow points. **j**, Multi-view factorization factor 1 variance explained across cell-type pairs (views). Abbreviations include cardiomyocytes (CM), endothelial cells (EN), FB, MY, pericytes (PC) and vascular smooth muscle cells (VM). **k**, To link deregulated inter- and intracellular signalling events, we combine ligand–receptor prior knowledge with differential contrast statistics between ischaemic samples versus the rest. Then, using knowledge of intracellular protein–protein interactions and TF regulons, we identify sign-coherent subnetworks that connect intercellular interactions (start nodes) with deregulated TFs (end nodes; Methods). **l**, A subset of interactions, the ligand and/or receptors of which are known to play a role in fibrosis[27,53,55] and were deregulated in FB and/or MY types. Only interactions with the highest loadings (>95th percentile) from the NMF analysis on spatially informed local ligand–receptor interactions were included in the differential expression analysis. Asterisks signify genes with FDR <0.05, and the colour bar corresponds to log2FC[47,48]. **m**, Sign-coherent signalling network originating from FN1 and SPP1 and propagating down to TFs deregulated in MY in ischaemia. **n**, Corresponding legend for the network in **m**. Source numerical data are available in source data.

These results underscore how LIANA+ offers a complete suite to identify disease-related communication patterns along with diverse strategies to interpret the underlying biological processes and intracellular signalling mechanisms.

## Discussion

In this work, we introduce LIANA+, which unifies and expands previous methodological developments, redefining them into synergistic components to enable diverse CCC analyses from single-cell and spatial (multi-)omics data.

Dissociated and spatially resolved data generation has commonly focused on transcriptomics[2,3], limiting CCC method development to protein-mediated interactions. Yet emerging multi-omics technologies enable the quantification of diverse molecules[15,26,58]. Building on a flexible and efficient infrastructure[28] and a rich biological knowledge base[22,59], we combined spatially informed multi-view modelling with local spatial metrics to examine the interactions of metabolites, receptors and cell types in a murine Parkinson's disease model[26]. Our analysis revealed dopamine-mediated interactions in specific brain subregions, highlighting how LIANA+ addresses challenges to integrate different data modalities, a scenario we anticipate to grow with the increasing combined use of complementary spatial technologies[15,39].

Single-cell technologies capture cellular heterogeneity but typically lose tissue architecture information during dissociation, leading to many false positives in spatially agnostic CCC inference, as our evaluations demonstrate. Using LIANA+, we jointly analysed the CCC events following myocardial infarction from single-nucleus and spatially resolved transcriptomics data[27], thereby addressing the resolution limitations of spatial transcriptomics data[15,39] and the anticipated high false positive rates from dissociated data. By combining NMF with spatially informed local interactions predicted from spatial transcriptomics data, we highlighted known drivers of fibrosis in ischaemic regions, such as SPP1 (refs. 27,53) and FN1 (refs. 53,55). After constraining the inference of CCC from dissociated transcriptomics data to co-localized cell types, ligands and receptors, we used a multi-view factorization approach[43] to identify FB as major sources of CCC events in ischaemia, potentially via a previously reported FB-to-MY signalling axis[27]. A targeted analysis of CCC and associated downstream signalling events revealed a causal signalling pathway linking FN1/SPP1 ligands to downstream TGF-β-associated TFs, such as SMAD3 and FOXO3 (refs. 56,57). Thus, this application highlights the potential of LIANA+ to generate unsupervised hypotheses and translate those into mechanistic biological insights.

The components of LIANA+ can be flexibly used in different ways beyond those in the illustrated examples. In particular, while we combined unsupervised factorizations with targeted differential analysis to uncover deregulated interactions, each approach can be used independently, depending on the dataset at hand. For example, if a specific hypothesis is available a priori, we may directly use a targeted analysis, while the unsupervised analyses might be better suited for complex experimental designs. Additionally, LIANA+ facilitates the flexible construction of models, allowing for the integration of an arbitrary number of modalities, as illustrated by our application of multi-view modelling[4] to study the relationships among metabolites, cell types and receptors. Consequently, with the increasing prevalence of spatial technologies that capture diverse molecular types[39], LIANA+ is well suited to study signalling events mediated by these molecules across combinations of modalities and technologies. Furthermore, LIANA+ enables the inference of putative causal signalling networks. To our knowledge, it is currently the only approach that considers the sign of the molecular interactions underlying inter- and intracellular signalling, a key feature of signal transduction. We applied this method to link deregulated protein-mediated CCC and TFs, but it should be applicable to any set of molecules, including metabolite-mediated signalling networks[60]. Finally, the flexibility of LIANA+ enables other

CCC methods, factorizations[61–63] or network approaches[64–66] to be incorporated.

The methods implemented in LIANA+ have a number of limitations. First, while each of its methods can flexibly infer interactions between any set of variables, they typically use prior knowledge, which is limited, often exhibiting biases and a trade-off between coverage and quality[3,24,67]. Most curation efforts have focused on annotating ligand–receptor interactions[19,21], and additional prior knowledge efforts are needed in particular for the inference of CCC beyond protein-mediated events. Moreover, contextualizing prior knowledge to specific cell types, tissues or diseases can help to reduce erroneous predictions. As an example, we customized MetalinksDB[68], a comprehensive resource for the inference of metabolite-mediated CCC, to brain-specific metabolites. Second, CCC from dissociated single-cell data remains limited to the co-expression of communication partners, which may not translate to the protein level, let alone imply a functional interaction[3]. Likewise, while spatially resolved data is a step further from its dissociated counterparts, it is limited to the co-localization of molecules. Finally, while our preliminary evaluations support the ability of LIANA+ to generate CCC insights across a range of technologies, systematic benchmarks of CCC methods are still pending. Ours and other evaluations remain limited by the lack of ground truth[3], using instead orthogonal modalities, such as spatial data[3,69] or downstream signalling[3]. As emerging technologies[70–72] that capture bona fide CCC events become measurable at scale and widely available, LIANA+ will facilitate such benchmarks and comparisons. In the meantime, LIANA+—like other CCC inference methods—remains a tool for hypothesis generation.

As illustrated in this manuscript, the synergistic components in LIANA+ can be combined in various ways, and their configurations can be tailored to address diverse and emerging questions and datasets. Given its modularity, new methods can be integrated into the framework and benefit from the established ecosystem of methods and resources. We further envision LIANA+ to be a versatile tool for the study of CCC driven by diverse mediators, beyond protein-mediated and metabolite-mediated interactions, expanding the range of CCC events that could be studied, such as host–microbiome interactions[73–75]. Thus, LIANA+ not only stands as a comprehensive and scalable tool for studying communication events but also serves as a catalyst for future developments in the field.

## Online content

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

# Methods

## Bivariate spatially informed metrics

LIANA+ implements spatially weighted global and local metrics to estimate bivariate co-localizations from spatial data. For statistical testing of the local metrics, we use spot label permutations to generate a null distribution against which empirical local $P$ values are computed. Inspired by GeoDa[82], we also categorize local bivariate associations according to the magnitude and sign of the two variables. We describe these in detail in Supplementary Note 3. When working with interactions, the members of which contain heteromeric complexes, we consider the minimum expression of complex members per spot. For any interactions where the members are not expressed in at least 10% of spots are excluded by default.

We provide a detailed tutorial on the bivariate metrics at https://liana-py.readthedocs.io/en/latest/notebooks/bivariate.html.

## Learning spatial relationships across multi-views

To learn multivariate interactions from spatially resolved data, we adapted MISTy's multi-view learning approach[4], where a view is a collection of variables (for example, a modality). This approach jointly models different spatial and functional aspects of the data, such that it can fit any number of views and each view can contain any number of variables. As shown in this work, one can use it to model different combinations of RNA expression, cell-type proportions and metabolite peak intensities.

In LIANA+, multi-view objects are represented as subclasses of MuData[77], modified to ensure the correct format of the views and corresponding spatial connectivities. Each multi-view object has an intrinsic view (intraview) that contains the target variables of interest for each observation. The other views can be considered as 'extra views', composed solely of predictor variables. Extra views can also represent different transformations of the variables within the intraview, for example, incorporating different spatial contexts[4]. Once the multi-view structure (model design) is defined, each target is modelled by predictors from each view independently. As such, for each target we obtain (1) relationship statistics ('importances') for each of the predictors from the distinct views, (2) the relative 'contribution' of each view to the joint prediction of each target and (3) the goodness of fit (for example, $R^2$) of the model. The statistics for each predictor (1) signifying its importance in the prediction of a given target variable, are calculated depending on the modelling approach. For example, for random forests, we use the reduction of variance explained that can be attributed to each predictor across all regression trees. For linear models, which were used throughout this manuscript, we calculate the ordinary least squares $t$-statistics of the estimated parameters under the zero value null hypothesis[83]. The independent view-specific predictions are combined by a cross-validated linear meta-model[4] (by default $k = 10$) to obtain the contributions of view-specific models (2), along with the goodness of fit of the overall model, for each target variable (3). In particular, we can discern between the contribution of the intraview, modelled as the intrinsic variability among target variables within the same cells/spots, from the joint predictive contribution of extra views, which encode spatial information.

By default, models are based on random forests[84] and can capture complex non-linear relationships. Here, we also implemented linear models as a signed alternative that provides the direction of relationships, though LIANA+ also accepts external single-view models.

To enable region-specific CCC modelling, we mask each extra view on the basis of labels (corresponding to regions) assigned to observations within the 'intra' view. This process is preceded by the spatial weighting of each extra view.

To facilitate the use of our multi-view learning approach, we provide in-depth tutorials using custom and predefined multi-view model structures at https://liana-py.readthedocs.io/en/latest/notebooks/misty.html.

## Estimation of spatial connectivities

As in MISTy[4], spatial connectivity weights are calculated using families of radial basis kernels: $w_{ij} = e^{-\frac{d_{ij}^2}{l^2}}$, Gaussian $w_{ij} = e^{-\frac{d_{ij}^2}{2l^2}}$, linear $w_{ij} = 1 - \frac{d_{ij}}{l}$ and exponential kernels $w_{ij} = e^{-\frac{d_{ij}}{l}}$; where $w$ is a weight matrix $w_{ij} \in [0,1]$ of shape $n \times n$; $d_{ij}$ is the Euclidean distance between cells or spots $i$ and $j$; and $l$ is a parameter controlling the length or bandwidth. We additionally use a cut-off parameter below which spatial connectivities are set to zero.

Throughout the manuscript, unless otherwise specified, we used Gaussian weights with a bandwidth of 150 and a cut-off of 0.1, and the diagonal (self to self) was set to 1 for the local scores. For multi-view learning, the diagonal was set to 0 (by default) to avoid self-referencing. For both the product and normalized product, we additionally apply L1 normalization to the weights. This adjustment accounts for the variability in the number of neighbours across different spots, implicitly accounted for by the remainder of the local metrics.

When working with multi-modal spatial technologies, the different modalities of which have observations with distinct locations, spatial connectivity weights are estimated according to a reference coordinate system. We detail this process in Supplementary Note 9.

## Ligand–receptor pathway enrichment

To perform ligand–receptor pathway enrichment, we first convert gene set (pathway) resources, represented as weighted bipartite graphs where each gene belongs to a gene set, into ligand–receptor sets. Specifically, we assign a weight to each ligand–receptor interaction on the basis of the mean weight of the ligands and receptors involved in the interaction, also taking into account the presence of heteromeric subunits. Moreover, we assign a given ligand–receptor interaction to a specific gene set (or pathway) only if all members of the interaction are part of the gene set, and in the case of weighted resources if all members are additionally sign-consistent. Finally, once a ligand–receptor resource is generated, we use decoupler-py to perform enrichment with univariate linear regression[81]. In this manuscript, we used the PROGENy resource[50] to assign pathway annotations to ligand–receptor interactions. In contrast to classic pathway gene sets, PROGENy contains consensually regulated targets of pathway perturbations, not genes thought to be members of the pathways. However, this resource-conversion procedure is applicable to any resource, including undirected resources, such as Gene Ontology terms for which all members of a gene set will be assigned a weight of 1.

## Hypothesis testing for deregulated CCC across conditions

To enable hypothesis testing for CCC, similarly to the strategy implemented in MultiNicheNet[16], we first generate pseudobulk profiles by summing raw expression counts for each sample and cell type with the decoupler-py package[81]. After filtering low-quality genes (for example, considering minimum expression in terms of total counts and samples in which the gene is expressed), we perform differential analysis for each cell type independently with DESeq2 (ref. 47), as implemented in PyDESeq2 (ref. 48).

Once feature statistics per cell type are generated, we transform those into a data frame of interaction statistics by joining them to a ligand–receptor resource, while additionally calculating average feature expression and expression proportions per cell type on the basis of a user-provided AnnData object[78]. Similarly to any other method in LIANA+, interactions expressed in less than 10% (by default) of the cells per cell type are filtered, considering the individual members of heteromeric complexes. A detailed tutorial is available at: https://liana-py.readthedocs.io/en/latest/notebooks/targeted.html.

## Sign-consistent intracellular networks

By combining CCC predictions with prior knowledge networks of intracellular signalling, here we uncover putative causal networks

linking CCC events to TFs. To accomplish this, we used CORNETO[79]—a Python package that unifies network inference problems from prior knowledge—to implement a modified version of the integer linear programming formulation implemented in CARNIVAL[60], described in Supplementary Note 10.

We show the inference of sign-consistent networks downstream of deregulated CCC events, identified using differential expression analysis with PyDESeq2[48] at https://liana-py.readthedocs.io/en/latest/notebooks/targeted.html.

## NMF on ligand–receptor local scores

A utility function was implemented that takes an AnnData object[78] as input and uses Scikit-learn's[84] NMF implementation to decompose the input matrix into two matrices of dimensions $k$, $n$ and $k$, $d$; where $d$ is the number of features, $n$ is the number of observations (cells) and $k$ is the number of components (factors). To estimate $k$, we additionally provide a heuristic elbow selection procedure, in which the optimal component number ($k$) is chosen from a sequential range of components using elbow selection as implemented in the kneedle package[85]. The selection of the optimal $k$ is based on the mean absolute reconstruction error.

## Identifying intercellular programmes across samples

Inspired by the CCC factorization approach proposed in Tensor-cell-2cell[20] and building on our recent application of multi-view factor analysis[43] to dissociated cross-condition atlases[86], we use the ligand–receptor inference methods from LIANA+ to infer interactions across each sample independently and then transform this into a multi-view structure of cell-type pairs (views), each represented by samples and the ligand–receptor interaction scores. To build the multi-view structure, we use the MuData format[77], and only views with at least 20 (by default) interactions in at least three (by default) samples are kept. Moreover, we exclude samples if they have less than ten interactions (by default), and interactions are considered only if they are present in at least 50% of the samples (by default). Then we use the Bayesian multi-view factor analysis approach, implemented within the MOFA+ statistical framework[43], to decompose ligand–receptor scores across samples into intercellular communication programmes. Specifically, the outputs of this factorization include a number of factors, each with (1) factor scores per sample, (2) ligand–receptor interaction factor loadings for each cell-type pair (view) and (3) variance explained per view (cell-type pair) for each factor. Tutorials on extracting intercellular programmes from single-cell dissociated data with multi-view factorization and Tensor-cell2cell are available at https://liana-py.readthedocs.io/en/latest/notebooks/mofatalk.html; https://liana-py.readthedocs.io/en/latest/notebooks/liana_c2c.html.

## Spot calling evaluation using local metrics

To benchmark how well each local score in LIANA+ preserves biological information, we devised spot classification and regression tasks. In the spot classification task, we used four public breast cancer 10X Visium slides[76], with annotations labelled as malignant (containing 'cancer' in their annotation) or non-malignant spots (any other spot). For each slide, we calculated ligand–receptor scores using the local metrics in LIANA+. Then for each local metric, we trained and evaluated random forest classifiers, with 100 estimators, using a stratified $K$-fold cross-validation strategy ($k = 10$). Area under the receiver operating characteristic curve (AUROC) and weighted F1 were calculated on the test sets, and their average across the folds was used in visualizations.

In the regression task, we used a public dataset with 28 10X Visium slides from left-ventricle heart tissues to compare how well different local metrics capture cell-type specific ligand–receptor events. In particular, we checked how well the local scores LIANA+ predict cell-type proportions per spot, inferred using cell2location[42] as done in

Kuppe et al.[27]. We used a random forest regressor, with 100 estimators, utilising a $K$-fold cross-validation training strategy ($k = 5$), and calculated the variance explained ($R^2$) and root mean squared error for each score. All classification and regression tasks were performed using Scikit-learn (v.1.3.2).

For the inference of ligand–receptor interactions throughout this work, we used LIANA's consensus resource—a resource combining the curated ligand–receptor resources in OmniPath[22].

## Spatial co-localization evaluation

To evaluate the agreement of spatially uniformed ligand–receptor methods in LIANA+ with spatially proximal ligand–receptor and cell-type pairs, we used five processed and recently published spatially informed single nucleus RNA sequencing (slide-tags) datasets[41]. Making use of spatial information for each dataset, we estimated global Moran's $R$ using LIANA+ to generate an indirect ground truth—that is, ligand–receptors and cell types co-localized to a greater extend than random (Moran's $R > 0$ and FDR $< 0.05$). Then we ran all ligand–receptor methods in LIANA+ without taking spatial information into account. AUROC was calculated for the whole distribution of each ligand–receptor method's scoring functions. To calculate balanced accuracy and normalized F1 (below), we used false positive filtering thresholds as suggested by each of the methods' authors (if available). For CellPhoneDB, CellChat and Geometric mean, interactions with $P$ values below 0.05 were filtered. For CellChat's ligand–receptor (LR) probabilities, we additionally only kept interactions for which either the ligand or receptor $P$ values were under 0.05. Similarly, for Connectome and log2 fold changes (log2FC), interactions were kept only if both ligand and receptor $P$ values were under 0.05 and had a positive scaled weight or log-fold change, respectively. For SingleCellSignalR, we considered ligand–receptor interactions with LR scores above 0.6. For LIANA's rank aggregate, we kept only those with a magnitude rank $< 0.05$. Since the authors of NATMI and scSeqComm do not suggest a threshold, we kept interactions if they were within the top 5% of the specificity weight and interscore distributions, respectively. Similarly, when evaluating the individual scoring functions from each method, we considered only the top 5% as positive predictions.

$$\text{Normalized F1} = \frac{1}{2}\left(\frac{\text{F1}_{observed}}{\text{F1}_{permuted}}\right)$$

where 'observed' is the F1 score for the actual ligand–receptor predictions, while the 'permuted' F1 was generated by shuffling the predictions of each method (100 times) and calculating an F1 score.

$$\text{Balanced accuracy} = \frac{1}{2}\left(\frac{\text{TP}}{\text{TP} + \text{FN}} + \frac{\text{TN}}{\text{TN} + \text{FP}}\right)$$

where TP is for true positives, TN is for true negatives, FP is for false positives and FN is for false negatives.

## Sample label classification

For the condition classification task, building on a similar approach[20], we used public, pre-processed, cross-conditional atlases (Supplementary Table 4), each selected such that they include more than five samples per condition following pre-processing. To ensure that only high-quality samples were used in each of the atlases, we removed any samples with less than 1,000 cells or $z$-transformed total counts above or below a $z$-score of 3 and −2, respectively. In the Carraro et al. dataset[87], we kept samples with more than 700 cells. Moreover, only cell types found in at least five samples and with at least 20 cells in each sample were considered. To ensure that the samples were balanced between the conditions if either condition had a sample ratio higher than 1.5 times the number of samples in the other condition, the over-represented condition was subsampled to the number of samples in

the underrepresented one. Each dataset was normalized to 10,000 total counts per cell and log1p-transformed.

Subsequent to pre-processing, we inferred ligand–receptor interactions at the cell-type level using the re-implemented (and homogenized) methods in LIANA+, independently for each sample. Any interactions not expressed in at least 10% of the cells in both source and receiver cell types were filtered.

Then the output from LIANA+ was converted to the structures used by the factorization approaches employed by MOFA+ and Tensor-cell-2cell—a multi-view object and a four-dimensional tensor, respectively. To run both factorization approaches, we considered interactions only if they were present in 33% of the samples, and any interactions missing in a sample were assumed to be biologically meaningful and assigned as zero. For all datasets, we decomposed the CCC events into 10 factors, except Reichart et al.[88], which was decomposed into 20 factors due to its larger sample size. Using the factor scores from each method–factorization combination we then performed a classification task, similar to the one from Armingol et al.[20]. Specifically, a random forest classifier, with 100 estimators, was trained and evaluated on the sample factor scores computed for each score-factorization combination, utilizing a stratified $K$-fold cross-validation strategy ($k = 3$), performed over five seeds. Then the mean AUROC and weighted F1 scores were calculated on the testing set's probabilities and label predictions, respectively.

## Prediction reliability in public heart failure atlases

To quantify the reliability of predictions from LIANA+, we evaluated the ability of different predictions to separate heart-failure versus non-heart-failure myeloid samples using six publicly available heart-failure datasets (Supplementary Table 5). Specifically, we used CCC predictions across different stages of the analysis of human myocardial infarction data[27] (Fig. 5), essentially progressively reducing the number of features (genes) to only those that were predicted as most relevant (Extended Data Fig. 7). Across each stage, predictions were chosen such that the resulting unique features (that is, ligand or receptor genes) were roughly equal to 25 (±1). For stages 1 and I (Extended Data Fig. 7a), we used the lowest CellPhoneDB $P$ values for MY and the highest Global Moran's $R$ calculated across all samples from the single-cell and spatial data, respectively. Then we generated myeloid pseudobulks from each sample and using the reduced features from each stage we calculated adjusted Rand index and silhouette scores with respect to the condition labels of the samples. To calculate the adjusted Rand index we used $k$-means clustering on the $z$-transformed normalized counts with $k = 2$. To estimate $z$-scores and $P$ values for the metrics across datasets, we generated null distributions using 1,000 randomly chosen sets of genes with the same size as the actual predictions.

## Analysis of murine Parkinson's disease model

We obtained a pre-processed dataset of three murine brain sections following 6-hydroxydopamine perturbation in one hemisphere[26], with joint metabolite and transcriptome measurements generated with matrix-assisted laser desorption/ionization–mass spectrometry imaging and 10X Visium technologies, respectively. We processed the metabolite and count matrices for each slide separately, applying standard log1p-normalization and standard quality control measures to the gene expression data. For the metabolite intensities, we used total count normalization followed by $z$-transformation. The observations of two modalities were manually aligned following the identification of tissue-containing observations for the metabolome data, using a procedure similar to the original publication[26]. We inferred cell-type proportions using Tangram's cell cluster level approach[35], fit with 1,000 epochs and a learning rate of 0.1. As a reference for deconvolution, we used an annotated single-cell dataset from Zeisel et al.[36]. Specifically, we used the 'TaxonomyRank4' cell group label, along with subgroups for dopaminergic and MSNs, which resulted in 48

refined murine brain cell types. Following the alignment of the two modalities, we modelled metabolite peaks (intraview) with cell-type proportions and brain-specific receptors as predictors (extra views). We bypassed modelling the intraview—that is, we did not model each metabolite peak by the remainder of the peaks (as done by default), since we were interested solely in the predictive performance of the extra views (receptors and cell types). We focused on the intersection of the top 250 highly variable metabolite peaks (targets) across the three slides and excluded any predictors with little-to-no variation—that is, genes not within the top 12,500 highly variable genes and cell types with a coefficient of variation below the 20th percentile. Brain-specific receptors were obtained from MetalinksDB[68], customized to include only metabolites found in the brain or cerebrospinal fluid. After the pre-processing steps, using our multi-view modelling procedure, we analysed 83 metabolite peaks as targets, along with 45 receptors and 37 cell types as predictors. Finally, we used spatially weighted cosine similarity on the $z$-transformed matrices of each modality to estimate the local scores and their corresponding $P$ values and categories. For all analyses, we used a bandwidth of 1,000 with a cut-off of 0.1 to calculate the spatial connectivities. A tutorial on spatially resolved multi-omics data integration is shown at https://liana-py.readthedocs.io/en/latest/notebooks/sma.html.

## Analysis of human myocardial infarction data

Following basic filtering and standard log1p-normalization, we estimated ligand–receptor local scores using spatially weighted cosine similarity on each of the 28 processed 10X Visium transcriptomics slides[27]. We considered interactions whose members were expressed in at least 10% of the spots. Then we concatenated the resulting ligand–receptor AnnData objects (slides) and kept only those interactions present in at least ten of the slides. Subsequently, we decomposed the concatenated object with NMF. We used an elbow selection procedure to automatically determine five factors as the optimal number. Then, we used Fisher's exact test to examine whether specific condition labels were enriched when considering samples with average factor scores above the 75th quantile. Pathway activities of ligand–receptor interaction loadings were calculated using linear regression[81] and sets of ligand–receptor pathways, annotated using the PROGENy pathway resource[50] with all genes. We used a pre-processed dataset of 29 single-nucleotide samples from the same publication[89]. Raw gene counts were normalized to 10,000 total counts per cell and log1p-transformed. We inferred ligand–receptor interactions per sample using LIANA's magnitude rank aggregate—a consensus of multiple magnitude-focused scores (Supplementary Table 3), considering only interactions with all members expressed in at least 5% of the cells per cell type. We further inferred ligand–receptor interactions only if they were deemed as condition relevant in the spatial analysis—that is, those with at least one standard deviation above the mean per NMF factor. Moreover, we considered cell-type pairs to interact only if they had a strong spatial relationship across all 10X Visium slides (target $R^2 > 0.05$; median $t$-value >1.645), as determined when the modelling of each cell-type proportion (inferred with cell2location[42]) by the remainder of the cell types. Then we decomposed the obtained ligand–receptor interactions from the dissociated data with multi-view factorization[43], considering interactions with at least 15 interactions in 20% of the samples, and views with at least 10 samples. Any missing interaction values were filled with zeroes.

For hypothesis testing, we generated pseudobulk profiles for cell type using decoupler-py[81], considering only genes with at least 10 counts across each of the samples or at least 20 counts in total, a large $n$ of 5, and expressed in at least 10% of the samples[81]. Then within each profile, we performed differential expression analysis with PyDeSeq2[47,48], contrasting ischaemic samples versus the rest—that is, we treated fibrotic and myogenic samples as baseline references. The output statistics were then converted into a data frame of ligand–receptor

differential statistics using LIANA+, keeping only interactions all members of which (including complex subunits) were expressed in at least 5% of the cells in source and target cell types.

Then we estimated TF activities using the Wald statistics from PyDeSeq2 with univariate linear regression[81] and CollecTRI[80]. For the inference of the downstream signalling events, we obtained OmniPath's protein–protein interaction network, considering interactions with consensus direction and a curation effort >3. Then using CORNETO (v0.9.1-alpha.5), we inferred the plausible causal networks propagating via the interaction between FN1/SPP1 and the ITGA5 and ITGB1 complex, down to all TFs identified as significantly deregulated (FDR <0.05) in MY. We used a gene expression proportion cut-off of 0.1, such that nodes above the cut-off were assigned a penalty of 1, and those below a penalty of 0.01. An edge penalty of 0.02 was also used, and to ensure the solution space was thoroughly explored, the problem was solved 100 times, each time introducing small amounts of uniform noise. We used the Gurobi[90] solver under an academic licence. Then the acyclic subnetworks obtained by each iteration were concatenated, such that edges from any solution were kept, and the network union was visualized with CytoScape[91].

### Statistics and reproducibility
The details for the pre-processing of the datasets used for the analyses are provided in the sections above. If not indicated otherwise in that section or the legends, no data were excluded from training and analysis. Similarly, unless stated otherwise, we used the default parameters for the methods within LIANA+ and any external method that we used. The details of statistical tests used throughout this study were provided in the corresponding method sections and the figure legends.

### Protocol
A step-by-step protocol for installing the software and an example application can be found on Nature Protocol Exchange[92].

### Reporting summary
Further information on research design is available in the Nature Portfolio Reporting Summary linked to this article.

### Data availability
Processed myocardial infarction single-nucleus and 10X Visium data were downloaded from the Human Cell Atlas (https://data.humancellatlas.org/explore/projects/e9f36305-d857-44a3-93f0-df4e6007dc97), also available via Zenodo at https://zenodo.org/records/6578047 (ref. 93). Processed breast cancer 10X Visium slides[76] (GSE176078) were obtained via https://zenodo.org/record/4739739 (ref. 94). Spatially resolved metabolome–transcriptome data[26] were obtained from https://data.mendeley.com/datasets/w7nw4km7xd/1, also available under GEO repository accession number GSE232910. Annotated single-cell mouse brain data[36], used as reference for deconvolution, were obtained from http://mousebrain.org/adolescent/, with GEO accession number GSE178265. Slide-tags datasets[41] were obtained via the Broad Institute Single Cell Portal: human brain−(SCP2167), mouse embryonic brain (SCP2170), mouse brain (SCP2162), human tonsil (SCP2169), human melanoma (SCP2171) and human melanoma multi-ome−SCP2176, also available under GEO GSE244355. All other data supporting the findings of this study are available as processed AnnData objects on Figshare (https://doi.org/10.6084/m9.figshare.26131789). All data used in this study are publicly available. Source data are provided with this paper.

### Code availability
LIANA+ is available via GitHub at https://github.com/saezlab/liana-py, along with detailed tutorials describing the distinct components presented here (https://liana-py.readthedocs.io). LIANA+ is available under a GPLv3 licence and is regularly released on GitHub with stable versions released on PyPI (https://pypi.org/project/liana/). The code for the analyses presented in this manuscript is available via GitHub at https://github.com/saezlab/lianaplus_manuscript. We used LIANA+ v1.0.5 release for the analyses presented here.

## References
76. Wu, S. Z. et al. A single-cell and spatially resolved atlas of human breast cancers. *Nat. Genet.* **53**, 1334–1347 (2021).

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

## Acknowledgements
D.D. was partially supported by the European Union's Horizon 2020 research and innovation program (no. 860329; Marie-Curie ITN 'STRATEGY-CKD'), S.L. by the European Union's Horizon 2020 research and innovation programme under grant agreement no. 965193 (DECIDER), P.S.L.S. from the Deutsche Forschungsgemeinschaft (DFG) under grant agreement SPP 2395, P.R.M. by the European Union's Horizon 2020 research and innovation programme under grant agreement no. 951773 (PerMedCoE) and R.O.R.F. by the DFG through CRC/SFB 1550 'Molecular Circuits of Heart Disease'. A.D. and P.B.M. are supported by funding from Pfizer and GSK, respectively, all through grants to J.S.R. We further thank A. Valdeolivas, E. Armingol, E. Kartal, O. Ivanova, M. Garrido-Rodriguez, H. Li, B. Humphreys, H. Baghdassarian, N. Steenbuck, J. Lundeberg, M. Vicari, G. Sturm, R. Becerra Perez, T. Rose and R. Fallegger for the helpful discussions. We are grateful to S. Müller-Dott for her help and critical input to the graphical abstract. We also thank G. Baruzzo, M. Baldan and B. Di Camillo for implementing scSeqComm in LIANA+.

## Author contributions

D.D. and J.S.R. conceived the project. D.D. developed the software, carried out the case studies and drafted the manuscript. P.S.L.S. and P.R.M. implemented the initial versions of the multi-view modelling and causal inference algorithms, respectively. E.F. and P.B-i-M. were involved in discussing the design and prototyping components related to the software. P.S.L.S., S.L., A.D., J.T. and R.O.R.F. provided feedback about the software, case studies and/or text, coordinated and integrated by D.D. J.S.R supervised the work. All authors approved the final manuscript.

## Competing interests

J.S.R. reports funding from GSK, Pfizer and Sanofi and fees/honoraria from Travere Therapeutics, Stadapharm, Astex, Pfizer, Owkin and Grunenthal. The remaining authors declare no competing interests.

## Additional information

**Extended data** is available for this paper at https://doi.org/10.1038/s41556-024-01469-w.

**Correspondence and requests for materials** should be addressed to Julio Saez-Rodriguez.

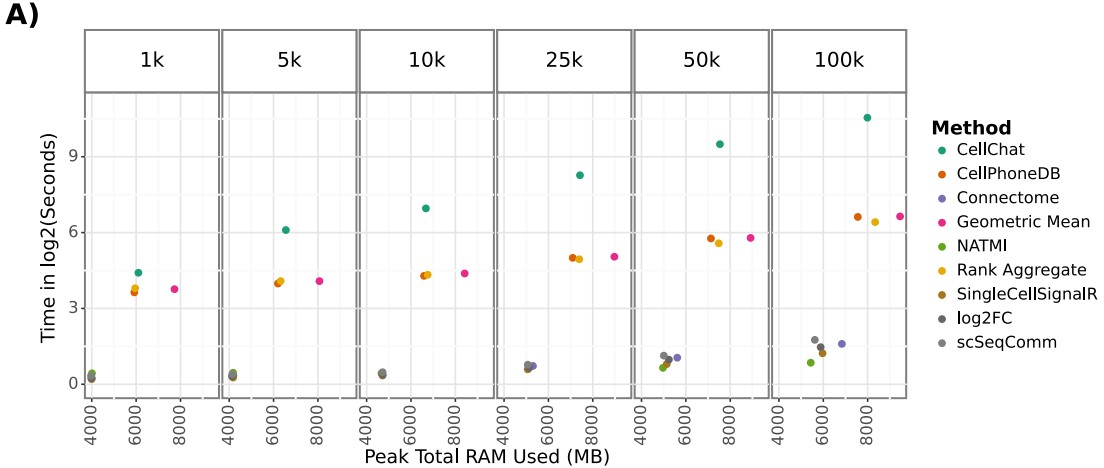

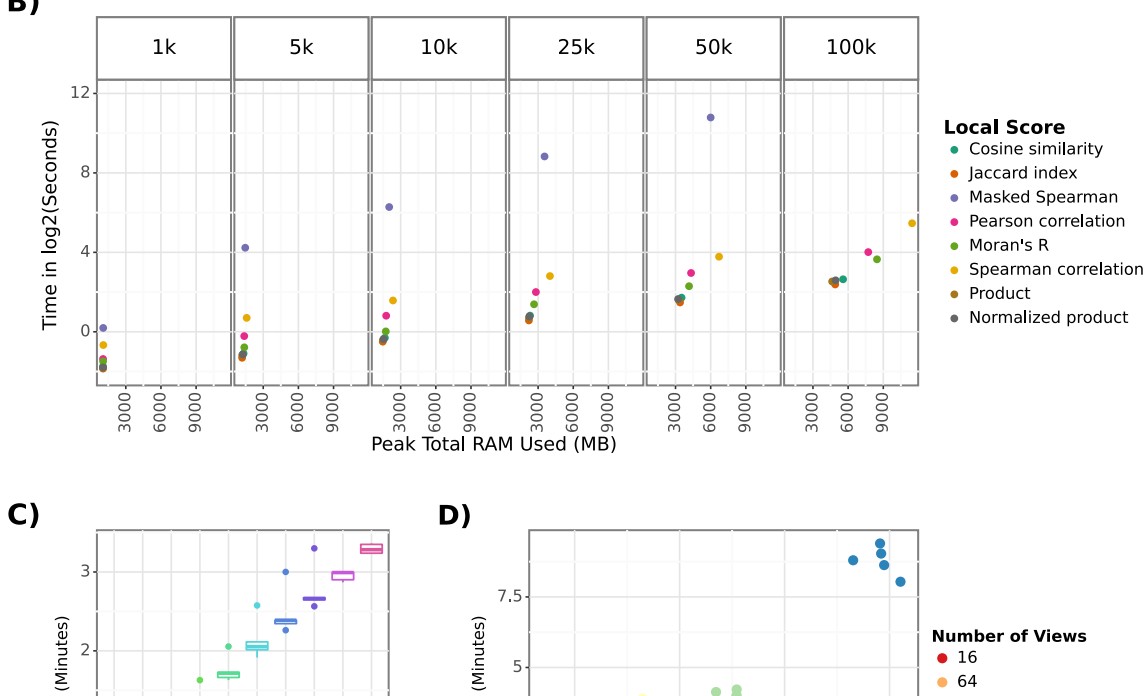

**Extended Data Fig. 1 | LIANA+ provides scalable implementations of cell−cell communication methods.** Efficiency benchmark of **A**) dissociated methods and **B**) spatially weighted local metrics implemented in LIANA + . Time in log2 of seconds and peak total RAM (memory) usage are shown over a number of simulated datasets with observations (cells) ranging from 1000 to 100,000 (1k to 100k). Masked Spearman Correlation was excluded for the 100k dataset.

**C**) Multi-view learning of cell-type spatial patterns across a range of views. RAM Usage is not shown as it was relatively constant at ~2,900MB ( ± 100MB) across views. **D**) Multi-view factor analysis used to identify intercellular programmes across a range of cell-type pairs (views). The median time in minutes and RAM usage are shown across five independent runs for each method. Source numerical data are available in source data.

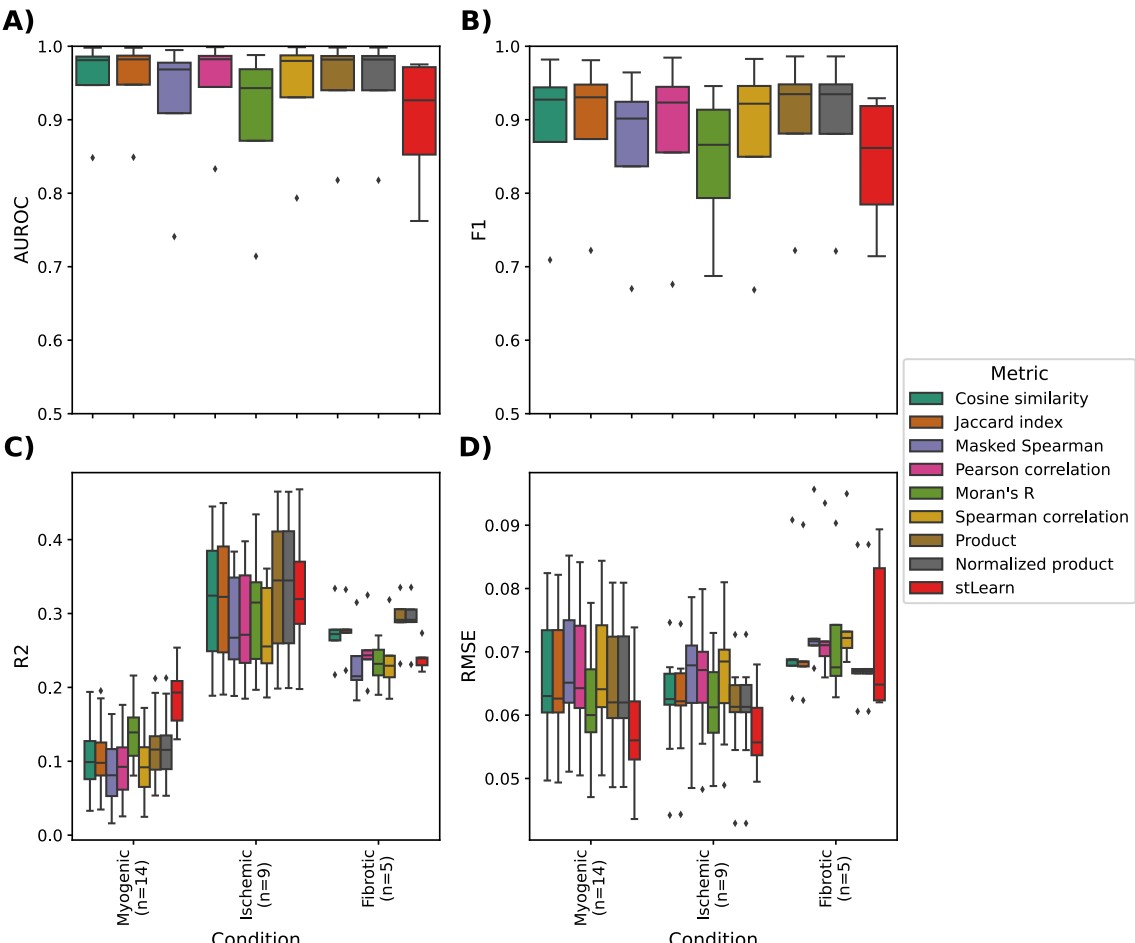

**Extended Data Fig. 2 | Spot-calling evaluations show that the local metrics in LIANA+ perform well at preserving biological information. A)** AUROC and **B)** weighted F1 when using local metrics to classify malignant spots in breast cancer spatial transcriptomics data (n = 4 samples)[76]; **C)** R2 and **D)** RMSE when using local metrics to predict cell type proportions in heart spatial transcriptomics

data[27]. The line in the boxplots represents the median with hinges showing the first and third quartiles and the whiskers extend up to 1.5 times the interquartile range above and below the box hinges. Outliers are shown as diamonds. Source numerical data are available in source data.

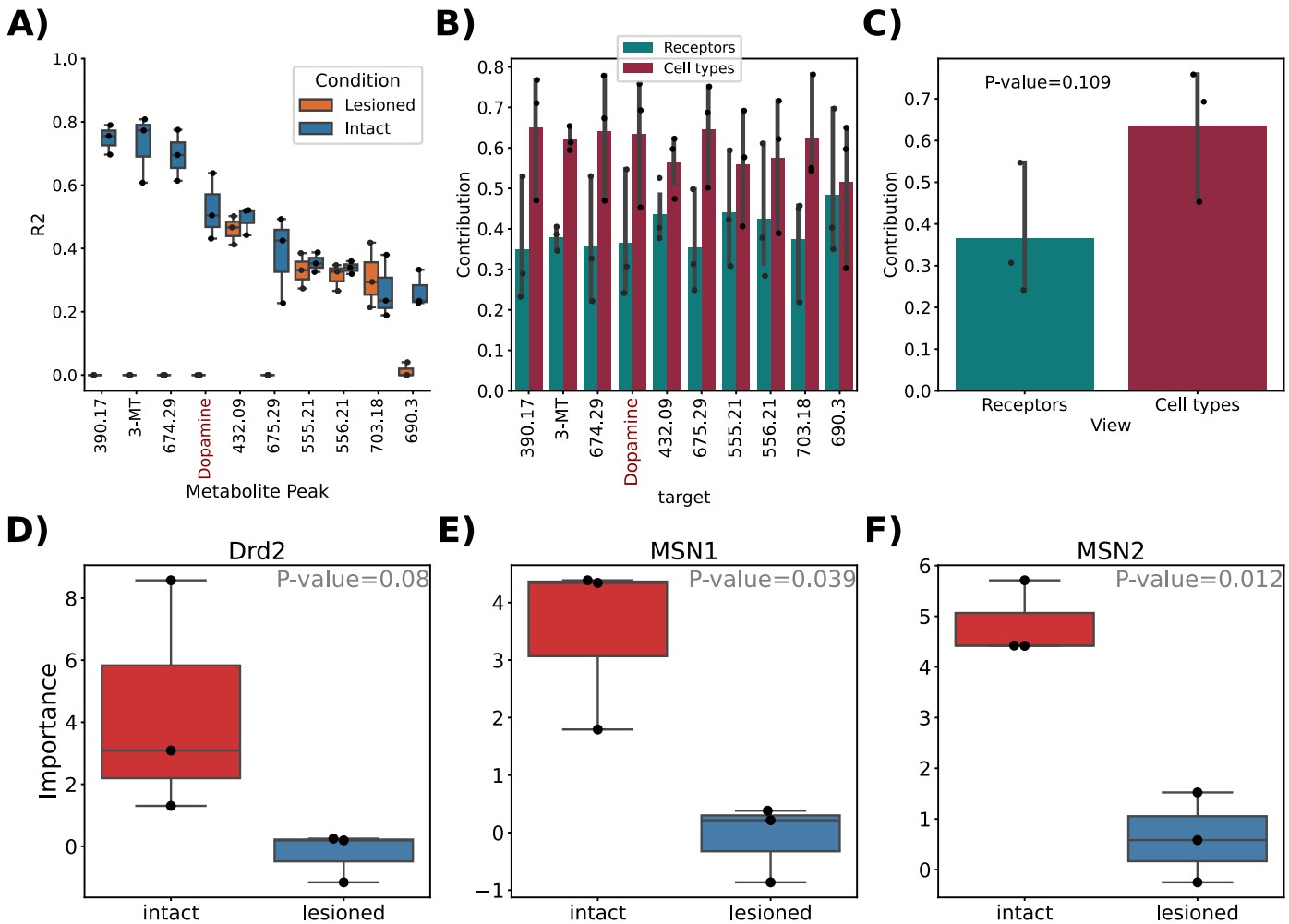

**Extended Data Fig. 3 | Multi-view modelling prediction performance per target in spatially resolved metabolite-transcriptome data. A)** Top 10 metabolite peaks with the highest variance explained (R2), including Dopamine (in dark red) and 3-methoxytyramine (3-MT) peaks, as annotated in the original publication[26]. **B)** Relative contributions of views (receptors and cell types) when jointly predicting metabolite peak intensities. **C)** Contributions of each view in predicting dopamine. Error bars represent 95% confidence intervals around the mean. The displayed P-value was calculated using a two-sided paired t-test across replicates (n = 3). **D-F)** Differences between lesioned and intact hemispheres in Dopamine's canonical Drd2 receptor, and medium MSN cell types 1 and 2. Uncorrected P-values were calculated using one-sided paired t-tests across replicates. The central line within each box marks the median, with the box hinges representing the first and third quartiles. The whiskers extend up to 1.5 times the interquartile range above and below the box hinges. Source numerical data are available in source data.

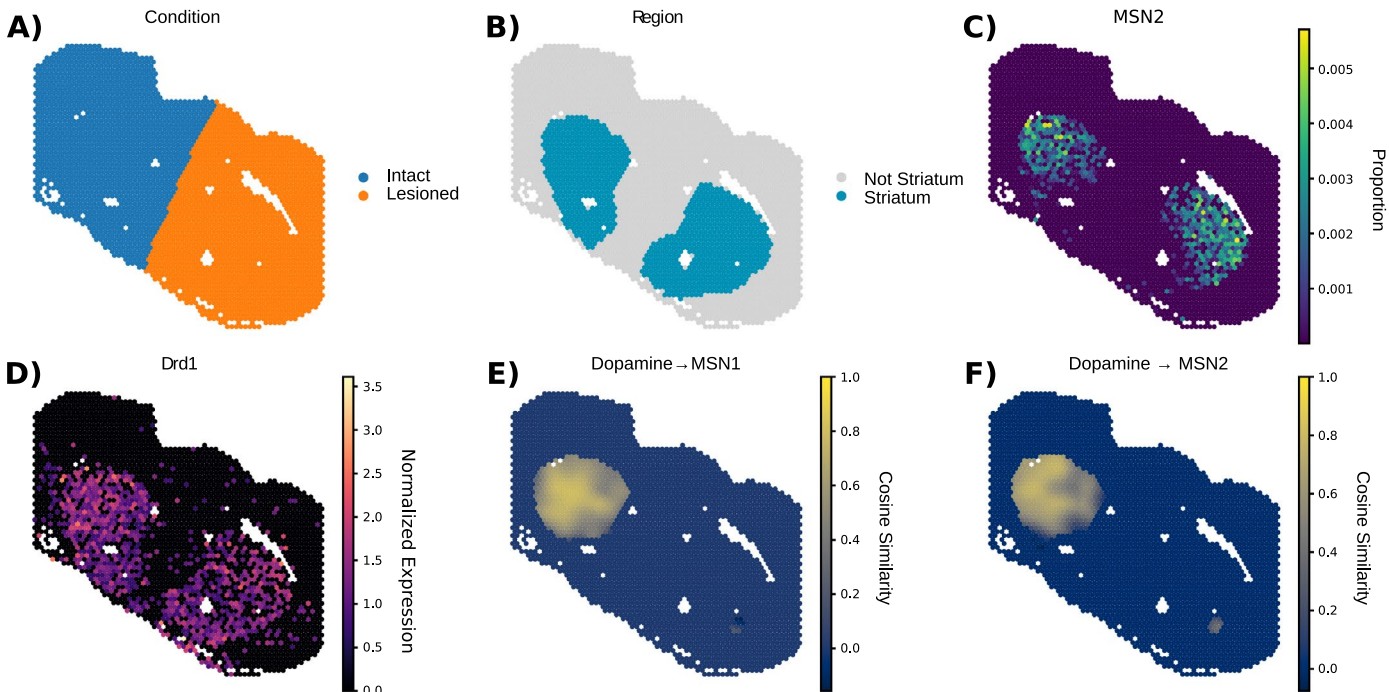

**Extended Data Fig. 4 | LIANA+ identifies local interactions from spatially resolved metabolite-transcriptome data. A)** Lesioned and Intact hemispheres; **B)** Striatum annotation; **C)** Medium Spiny Neuron (MSN) 2 cell type proportions, and **D)** *Drd1* expression. Interactions between Dopamine and MSNs **E)** 1 and **F)** 2, calculated using spatially-weighted cosine similarity. All panels show slide B1 from experiment V11L12-109.

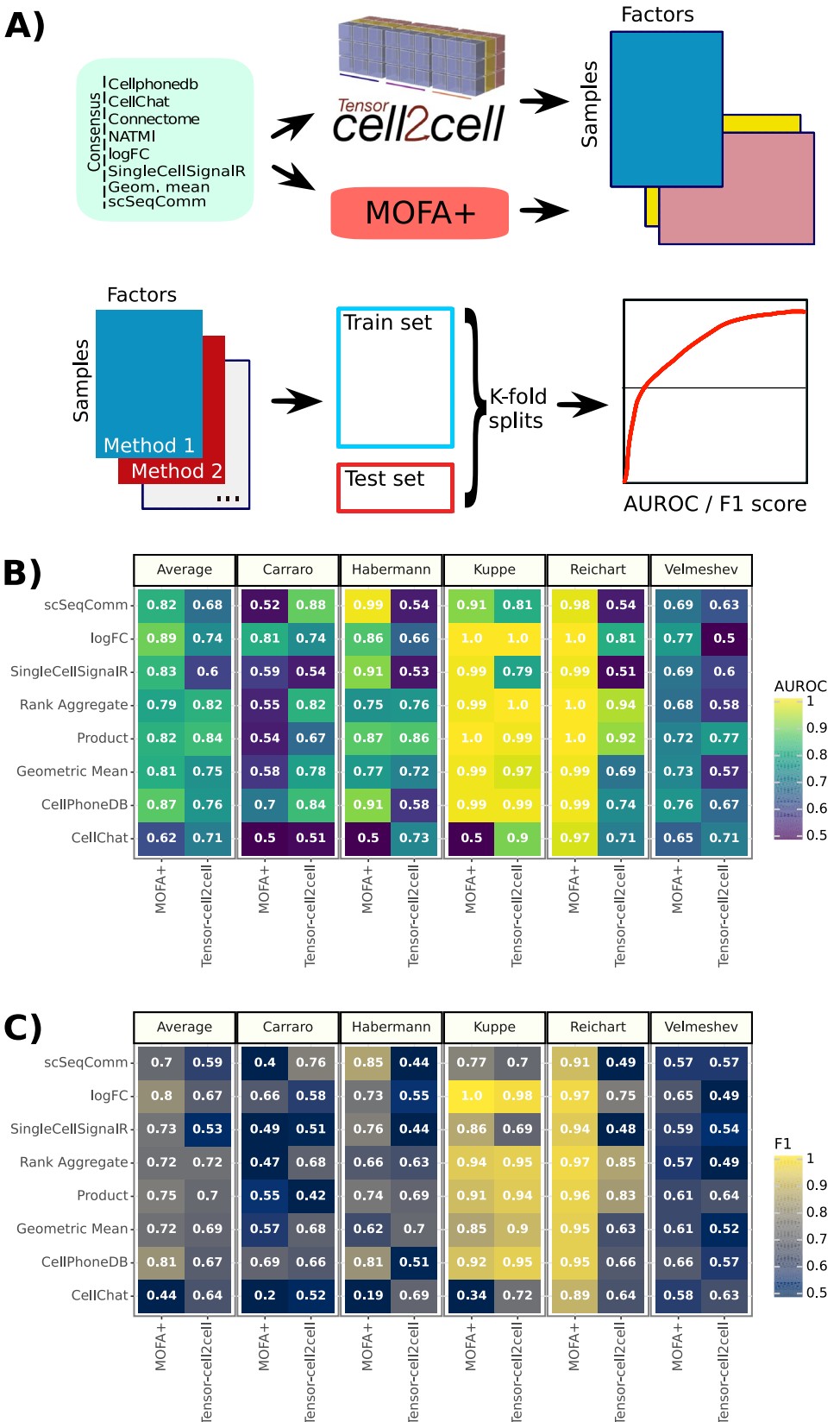

**Extended Data Fig. 5 | LIANA+ captures intercellular programmes that separate patient samples according to different conditions. A)** Classification setup to evaluate the ability of ligand-receptor methods, combined with Tensor-cell2cell and multi-view factor analysis (MOFA+), in separating conditions from multi-condition atlases (n = 5) in an unsupervised manner. **B)** Area under the receiver-operator curve (AUROC) and **C)** weighted F1 score, as calculated across 5 datasets, and the 'Average' performance. Source numerical data are available in source data.

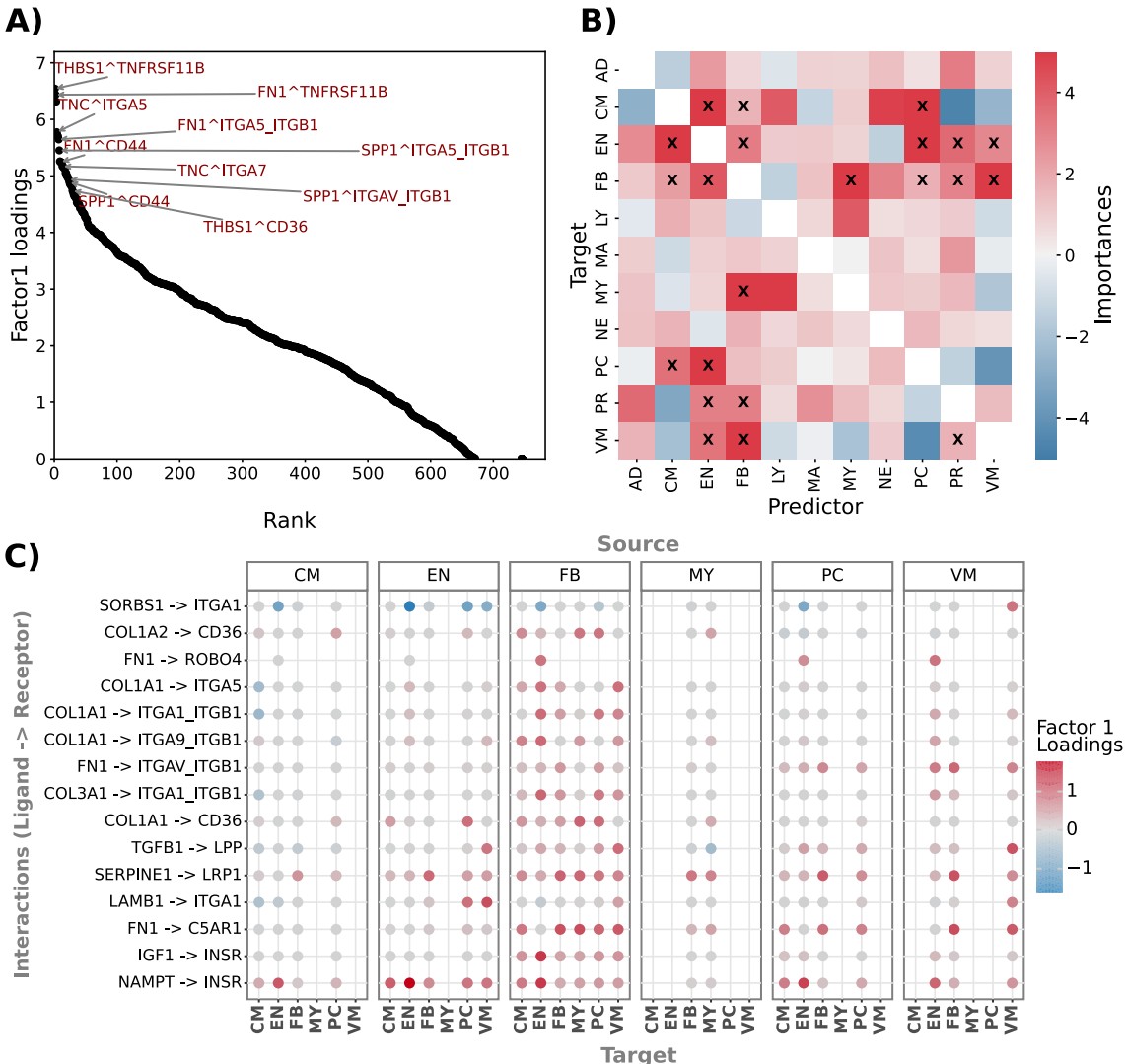

**Extended Data Fig. 6 | Condition-relevant ligand-receptor interactions and spatially-co-localized cell types in Myocardial infarction. A)** Top 30 interactions (with 'FN1', 'TNC', 'THBS1', and 'SPP1 as ligands) in Factor 1 identified using NMF on local ligand-receptor metrics in spatially resolved 10X Visium heart samples. **B)** Importances (Median t-values clipped at 5 and -5; n = 28) from a spatially-weighted model predicting all possible cell type interactions from Myocardial infarction 10X Visium slides. Cell type interactions with Median t-value > 1.645 and R2 > 5% are marked with X. **C)** Multi-view Factorization feature (interaction) loadings following decomposition of ligand-receptor scores inferred from dissociated single-nucleus myocardial infarction data. The top 15 interactions with the highest interaction loadings are shown across all cell-type pairs. Abbreviations used include AD for Adipocytes, CM for Cardiomyocytes, EN for Endothelial cells, FB for Fibroblasts, PC for Pericytes, VM for Vascular smooth muscle cells, NE for Neuronal cells, MY for Myeloid cells, MA for Mast cells, and LY for Lymphoid cells. Source numerical data are available in source data.

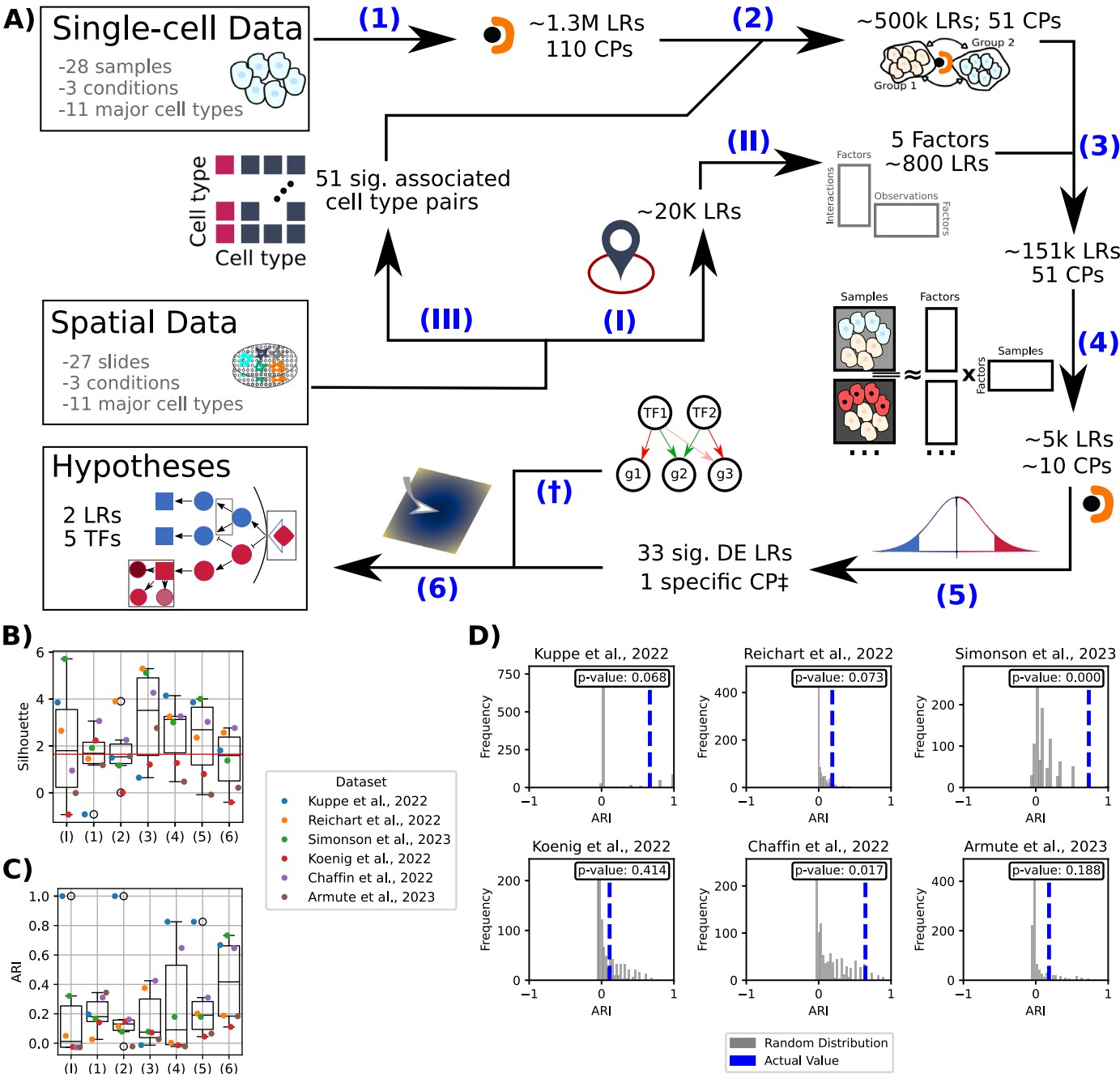

**Extended Data Fig. 7 | LIANA+ generates interpretable and reliable predictions from complex datasets. A)** Applying LIANA+ on Myocardial infarction substantially reduced the prediction space to yield an interpretable subset of predictions. (1) inference of the full possible prediction space of ligand-receptor interactions across 110 cell-type pairs, 28 samples, and 3 conditions. (2) Incorporation of spatially associated cell-type pairs (III). (3) Inclusion of information about spatially-colocalised ligand-receptor interactions (LRs), identified using (I) local scores with non-negative matrix factorization (II). (4) Multi-view factor analysis using LRs predicted from single-cell data and informed by the spatial analysis. (5) Targeted differential expression analysis to find significantly deregulated LRs in a specific pair of interacting literature-supported cell types (‡). (6) A causal network search to generate a specific mechanistic hypothesis linking LR interactions to downstream Transcription Factors (TFs). The TF activities were estimated using a curated TF-target network[80] (IV) and linked to the LRs using a protein–protein interaction network[22]. **B)** Mean silhouette scores of clustering heart failure versus healthy myeloid samples using genes extracted from CCC predictions across the distinct steps shown in subpanel **A**. Silhouette scores were z-transformed using randomly permuted gene sets of the same size as a baseline. The red dashed line signifies a z-score of 1.645, equivalent to a significant one-sided z-test. **C)** Adjusted Rand Index (ARI) across the distinct steps shown in subpanel **A**. The central line within each box marks the median, with the box hinges representing the first and third quartiles. The whiskers extend up to 1.5 times the interquartile range above and below the box hinges. Hollow points represent outliers. In the evaluation, Steps 1 and I were respectively represented by prediction averages from CellPhoneDB P-values and SpatialDM's Global Moran's R across samples (Methods). **D)** Histogram of ARI values for **Step 6**, with one-sided empirical P-values calculated against a random distribution generated using sets of genes of the same size as the predictions in each step. The grey boxes depict the frequency of values drawn at random; the blue dashed line corresponds to the actual value. Source numerical data are available in source data.

# Reporting Summary

## Statistics

For all statistical analyses, confirm that the following items are present in the figure legend, table legend, main text, or Methods section.

| n/a | Confirmed | |
|---|---|---|
| ☐ | ☒ | The exact sample size (*n*) for each experimental group/condition, given as a discrete number and unit of measurement |
| ☐ | ☒ | A statement on whether measurements were taken from distinct samples or whether the same sample was measured repeatedly |
| ☐ | ☒ | The statistical test(s) used AND whether they are one- or two-sided<br>*Only common tests should be described solely by name; describe more complex techniques in the Methods section.* |
| ☒ | ☐ | A description of all covariates tested |
| ☐ | ☒ | A description of any assumptions or corrections, such as tests of normality and adjustment for multiple comparisons |
| ☐ | ☒ | A full description of the statistical parameters including central tendency (e.g. means) or other basic estimates (e.g. regression coefficient) AND variation (e.g. standard deviation) or associated estimates of uncertainty (e.g. confidence intervals) |
| ☐ | ☒ | For null hypothesis testing, the test statistic (e.g. *F*, *t*, *r*) with confidence intervals, effect sizes, degrees of freedom and *P* value noted<br>*Give P values as exact values whenever suitable.* |
| ☐ | ☒ | For Bayesian analysis, information on the choice of priors and Markov chain Monte Carlo settings |
| ☒ | ☐ | For hierarchical and complex designs, identification of the appropriate level for tests and full reporting of outcomes |
| ☐ | ☒ | Estimates of effect sizes (e.g. Cohen's *d*, Pearson's *r*), indicating how they were calculated |

*Our web collection on statistics for biologists contains articles on many of the points above.*

## Software and code

Policy information about availability of computer code

| Data collection | The software for LIANA+ is available from https://github.com/saezlab/liana-py<br>The development release used for the paper is available here: https://github.com/saezlab/liana-py/releases/tag/1.0.5 |
|---|---|
| Data analysis | The data processing and analysis supporting the findings in the study can be reproduced using the scripts and notebooks available at: https://github.com/saezlab/lianaplus_manuscript<br>Software packages used for the analysis include:<br>LIANA+ (v1.0.5); numpy (v1.24.4); pandas (v2.0.3); muon (v0.1.5); scanpy (v1.9.8); scikit-learn (v1.3.2); squidpy (v1.3.1); Python (v3.10.13); cell2cell (v0.7.2); CellPhoneDBv2 was cached via OmniPath and is versioned in LIANA+; umap-learn (v0.5.5) |

For manuscripts utilizing custom algorithms or software that are central to the research but not yet described in published literature, software must be made available to editors and reviewers. We strongly encourage code deposition in a community repository (e.g. GitHub). See the Nature Portfolio guidelines for submitting code & software for further information.

## Data

Policy information about availability of data

All manuscripts must include a data availability statement. This statement should provide the following information, where applicable:

- Accession codes, unique identifiers, or web links for publicly available datasets
- A description of any restrictions on data availability
- For clinical datasets or third party data, please ensure that the statement adheres to our policy

Processed myocardial infarction single-nucleus and 10X Visium data was downloaded from the Human Cell Atlas (https://data.humancellatlas.org/explore/projects/e9f36305-d857-44a3-93f0-df4e6007dc97), also available via Zenodo at https://zenodo.org/records/6578047.
Processed breast cancer 10× Visium slides 81 (GSE176078; https://www.ncbi.nlm.nih.gov/geo/query/acc.cgi?acc=GSE176078) were obtained via https://zenodo.org/record/4739739.
spatially resolved metabolome-transcriptome data 26 was obtained from https://data.mendeley.com/datasets/w7nw4km7xd/1, also available under GEO repository accession number GSE232910. Annotated single-cell mouse brain data 36, used for reference for deconvolution, was obtained from http://mousebrain.org/adolescent/, with GEO accession number GSE178265.
Slide-tags datasets 41 were obtained via the Broad Institute Single Cell Portal: human brain - (SCP2167), mouse embryonic brain (SCP2170); mouse brain (SCP2162); human tonsil (SCP2169); human melanoma (SCP2171); and human melanoma multi-ome - SCP2176; also available under GEO GSE244355. Source data have been provided in Source Data. All other data supporting the findings of this study are available as processed AnnData objects on figshare (https://doi.org/10.6084/m9.figshare.26131789). All data used in this study is publicly available.

## Research involving human participants, their data, or biological material

Policy information about studies with human participants or human data. See also policy information about sex, gender (identity/presentation), and sexual orientation and race, ethnicity and racism.

| | |
|---|---|
| Reporting on sex and gender | N/A |
| Reporting on race, ethnicity, or other socially relevant groupings | N/A |
| Population characteristics | N/A |
| Recruitment | N/A |
| Ethics oversight | N/A |

Note that full information on the approval of the study protocol must also be provided in the manuscript.

# Field-specific reporting

Please select the one below that is the best fit for your research. If you are not sure, read the appropriate sections before making your selection.

☒ Life sciences  ☐ Behavioural & social sciences  ☐ Ecological, evolutionary & environmental sciences

For a reference copy of the document with all sections, see nature.com/documents/nr-reporting-summary-flat.pdf

# Life sciences study design

All studies must disclose on these points even when the disclosure is negative.

| | |
|---|---|
| Sample size | All datasets used in this paper are publicly available and were not generated for this study. The sample size for each study is reported in the figure captions. |
| Data exclusions | We only removed samples and features following standard procedures as described in the manuscript. |
| Replication | This is not relevant for this study as no wet lab experiments were performed. The replication and code for computational experiments is deposited at https://github.com/saezlab/lianaplus_manuscript |
| Randomization | This is not relevant for this study as no wet lab experiments were performed. |
| Blinding | This is not relevant for this study as no wet lab experiments were performed. |

# Behavioural & social sciences study design

All studies must disclose on these points even when the disclosure is negative.

| Study description | Briefly describe the study type including whether data are quantitative, qualitative, or mixed-methods (e.g. qualitative cross-sectional, quantitative experimental, mixed-methods case study). |
| --- | --- |
| Research sample | State the research sample (e.g. Harvard university undergraduates, villagers in rural India) and provide relevant demographic information (e.g. age, sex) and indicate whether the sample is representative. Provide a rationale for the study sample chosen. For studies involving existing datasets, please describe the dataset and source. |
| Sampling strategy | Describe the sampling procedure (e.g. random, snowball, stratified, convenience). Describe the statistical methods that were used to predetermine sample size OR if no sample-size calculation was performed, describe how sample sizes were chosen and provide a rationale for why these sample sizes are sufficient. For qualitative data, please indicate whether data saturation was considered, and what criteria were used to decide that no further sampling was needed. |
| Data collection | Provide details about the data collection procedure, including the instruments or devices used to record the data (e.g. pen and paper, computer, eye tracker, video or audio equipment) whether anyone was present besides the participant(s) and the researcher, and whether the researcher was blind to experimental condition and/or the study hypothesis during data collection. |
| Timing | Indicate the start and stop dates of data collection. If there is a gap between collection periods, state the dates for each sample cohort. |
| Data exclusions | If no data were excluded from the analyses, state so OR if data were excluded, provide the exact number of exclusions and the rationale behind them, indicating whether exclusion criteria were pre-established. |
| Non-participation | State how many participants dropped out/declined participation and the reason(s) given OR provide response rate OR state that no participants dropped out/declined participation. |
| Randomization | If participants were not allocated into experimental groups, state so OR describe how participants were allocated to groups, and if allocation was not random, describe how covariates were controlled. |

# Ecological, evolutionary & environmental sciences study design

All studies must disclose on these points even when the disclosure is negative.

| Study description | Briefly describe the study. For quantitative data include treatment factors and interactions, design structure (e.g. factorial, nested, hierarchical), nature and number of experimental units and replicates. |
| --- | --- |
| Research sample | Describe the research sample (e.g. a group of tagged Passer domesticus, all Stenocereus thurberi within Organ Pipe Cactus National Monument), and provide a rationale for the sample choice. When relevant, describe the organism taxa, source, sex, age range and any manipulations. State what population the sample is meant to represent when applicable. For studies involving existing datasets, describe the data and its source. |
| Sampling strategy | Note the sampling procedure. Describe the statistical methods that were used to predetermine sample size OR if no sample-size calculation was performed, describe how sample sizes were chosen and provide a rationale for why these sample sizes are sufficient. |
| Data collection | Describe the data collection procedure, including who recorded the data and how. |
| Timing and spatial scale | Indicate the start and stop dates of data collection, noting the frequency and periodicity of sampling and providing a rationale for these choices. If there is a gap between collection periods, state the dates for each sample cohort. Specify the spatial scale from which the data are taken |
| Data exclusions | If no data were excluded from the analyses, state so OR if data were excluded, describe the exclusions and the rationale behind them, indicating whether exclusion criteria were pre-established. |
| Reproducibility | Describe the measures taken to verify the reproducibility of experimental findings. For each experiment, note whether any attempts to repeat the experiment failed OR state that all attempts to repeat the experiment were successful. |
| Randomization | Describe how samples/organisms/participants were allocated into groups. If allocation was not random, describe how covariates were controlled. If this is not relevant to your study, explain why. |
| Blinding | Describe the extent of blinding used during data acquisition and analysis. If blinding was not possible, describe why OR explain why blinding was not relevant to your study. |

Did the study involve field work? ☐ Yes ☐ No

# Field work, collection and transport

| | |
|---|---|
| Field conditions | *Describe the study conditions for field work, providing relevant parameters (e.g. temperature, rainfall).* |
| Location | *State the location of the sampling or experiment, providing relevant parameters (e.g. latitude and longitude, elevation, water depth).* |
| Access & import/export | *Describe the efforts you have made to access habitats and to collect and import/export your samples in a responsible manner and in compliance with local, national and international laws, noting any permits that were obtained (give the name of the issuing authority, the date of issue, and any identifying information).* |
| Disturbance | *Describe any disturbance caused by the study and how it was minimized.* |

# Reporting for specific materials, systems and methods

We require information from authors about some types of materials, experimental systems and methods used in many studies. Here, indicate whether each material, system or method listed is relevant to your study. If you are not sure if a list item applies to your research, read the appropriate section before selecting a response.

## Materials & experimental systems

| n/a | Involved in the study |
|---|---|
| ☒ | ☐ Antibodies |
| ☒ | ☐ Eukaryotic cell lines |
| ☒ | ☐ Palaeontology and archaeology |
| ☒ | ☐ Animals and other organisms |
| ☒ | ☐ Clinical data |
| ☒ | ☐ Dual use research of concern |
| ☒ | ☐ Plants |

## Methods

| n/a | Involved in the study |
|---|---|
| ☒ | ☐ ChIP-seq |
| ☒ | ☐ Flow cytometry |
| ☒ | ☐ MRI-based neuroimaging |

## Antibodies

| | |
|---|---|
| Antibodies used | *Describe all antibodies used in the study; as applicable, provide supplier name, catalog number, clone name, and lot number.* |
| Validation | *Describe the validation of each primary antibody for the species and application, noting any validation statements on the manufacturer's website, relevant citations, antibody profiles in online databases, or data provided in the manuscript.* |

## Eukaryotic cell lines

Policy information about cell lines and Sex and Gender in Research

| | |
|---|---|
| Cell line source(s) | *State the source of each cell line used and the sex of all primary cell lines and cells derived from human participants or vertebrate models.* |
| Authentication | *Describe the authentication procedures for each cell line used OR declare that none of the cell lines used were authenticated.* |
| Mycoplasma contamination | *Confirm that all cell lines tested negative for mycoplasma contamination OR describe the results of the testing for mycoplasma contamination OR declare that the cell lines were not tested for mycoplasma contamination.* |
| Commonly misidentified lines (See ICLAC register) | *Name any commonly misidentified cell lines used in the study and provide a rationale for their use.* |

## Palaeontology and Archaeology

| | |
|---|---|
| Specimen provenance | *Provide provenance information for specimens and describe permits that were obtained for the work (including the name of the issuing authority, the date of issue, and any identifying information). Permits should encompass collection and, where applicable, export.* |
| Specimen deposition | *Indicate where the specimens have been deposited to permit free access by other researchers.* |

| Dating methods | *If new dates are provided, describe how they were obtained (e.g. collection, storage, sample pretreatment and measurement), where they were obtained (i.e. lab name), the calibration program and the protocol for quality assurance OR state that no new dates are provided.* |

☐ Tick this box to confirm that the raw and calibrated dates are available in the paper or in Supplementary Information.

| Ethics oversight | *Identify the organization(s) that approved or provided guidance on the study protocol, OR state that no ethical approval or guidance was required and explain why not.* |

Note that full information on the approval of the study protocol must also be provided in the manuscript.

# Animals and other research organisms

Policy information about studies involving animals; ARRIVE guidelines recommended for reporting animal research, and Sex and Gender in Research

| Laboratory animals | *For laboratory animals, report species, strain and age OR state that the study did not involve laboratory animals.* |
| Wild animals | *Provide details on animals observed in or captured in the field; report species and age where possible. Describe how animals were caught and transported and what happened to captive animals after the study (if killed, explain why and describe method; if released, say where and when) OR state that the study did not involve wild animals.* |
| Reporting on sex | *Indicate if findings apply to only one sex; describe whether sex was considered in study design, methods used for assigning sex. Provide data disaggregated for sex where this information has been collected in the source data as appropriate; provide overall numbers in this Reporting Summary. Please state if this information has not been collected. Report sex-based analyses where performed, justify reasons for lack of sex-based analysis.* |
| Field-collected samples | *For laboratory work with field-collected samples, describe all relevant parameters such as housing, maintenance, temperature, photoperiod and end-of-experiment protocol OR state that the study did not involve samples collected from the field.* |
| Ethics oversight | *Identify the organization(s) that approved or provided guidance on the study protocol, OR state that no ethical approval or guidance was required and explain why not.* |

Note that full information on the approval of the study protocol must also be provided in the manuscript.

# Clinical data

Policy information about clinical studies
All manuscripts should comply with the ICMJE guidelines for publication of clinical research and a completed CONSORT checklist must be included with all submissions.

| Clinical trial registration | *Provide the trial registration number from ClinicalTrials.gov or an equivalent agency.* |
| Study protocol | *Note where the full trial protocol can be accessed OR if not available, explain why.* |
| Data collection | *Describe the settings and locales of data collection, noting the time periods of recruitment and data collection.* |
| Outcomes | *Describe how you pre-defined primary and secondary outcome measures and how you assessed these measures.* |

# Dual use research of concern

Policy information about dual use research of concern

## Hazards

Could the accidental, deliberate or reckless misuse of agents or technologies generated in the work, or the application of information presented in the manuscript, pose a threat to:

No | Yes
☐ | ☐ Public health
☐ | ☐ National security
☐ | ☐ Crops and/or livestock
☐ | ☐ Ecosystems
☐ | ☐ Any other significant area

## Experiments of concern

Does the work involve any of these experiments of concern:

No | Yes
☐ | ☐ Demonstrate how to render a vaccine ineffective
☐ | ☐ Confer resistance to therapeutically useful antibiotics or antiviral agents
☐ | ☐ Enhance the virulence of a pathogen or render a nonpathogen virulent
☐ | ☐ Increase transmissibility of a pathogen
☐ | ☐ Alter the host range of a pathogen
☐ | ☐ Enable evasion of diagnostic/detection modalities
☐ | ☐ Enable the weaponization of a biological agent or toxin
☐ | ☐ Any other potentially harmful combination of experiments and agents

# Plants

| | |
|---|---|
| Seed stocks | N/A |
| Novel plant genotypes | N/A |
| Authentication | N/A |

# ChIP-seq

## Data deposition

☐ Confirm that both raw and final processed data have been deposited in a public database such as GEO.

☐ Confirm that you have deposited or provided access to graph files (e.g. BED files) for the called peaks.

| | |
|---|---|
| Data access links<br>*May remain private before publication.* | *For "Initial submission" or "Revised version" documents, provide reviewer access links. For your "Final submission" document, provide a link to the deposited data.* |
| Files in database submission | *Provide a list of all files available in the database submission.* |
| Genome browser session<br>(e.g. UCSC) | *Provide a link to an anonymized genome browser session for "Initial submission" and "Revised version" documents only, to enable peer review. Write "no longer applicable" for "Final submission" documents.* |

## Methodology

| | |
|---|---|
| Replicates | *Describe the experimental replicates, specifying number, type and replicate agreement.* |
| Sequencing depth | *Describe the sequencing depth for each experiment, providing the total number of reads, uniquely mapped reads, length of reads and whether they were paired- or single-end.* |
| Antibodies | *Describe the antibodies used for the ChIP-seq experiments; as applicable, provide supplier name, catalog number, clone name, and lot number.* |
| Peak calling parameters | *Specify the command line program and parameters used for read mapping and peak calling, including the ChIP, control and index files used.* |
| Data quality | *Describe the methods used to ensure data quality in full detail, including how many peaks are at FDR 5% and above 5-fold enrichment.* |
| Software | *Describe the software used to collect and analyze the ChIP-seq data. For custom code that has been deposited into a community repository, provide accession details.* |

# Flow Cytometry

## Plots

Confirm that:

☐ The axis labels state the marker and fluorochrome used (e.g. CD4-FITC).

☐ The axis scales are clearly visible. Include numbers along axes only for bottom left plot of group (a 'group' is an analysis of identical markers).

☐ All plots are contour plots with outliers or pseudocolor plots.

☐ A numerical value for number of cells or percentage (with statistics) is provided.

## Methodology

| | |
|---|---|
| Sample preparation | *Describe the sample preparation, detailing the biological source of the cells and any tissue processing steps used.* |
| Instrument | *Identify the instrument used for data collection, specifying make and model number.* |
| Software | *Describe the software used to collect and analyze the flow cytometry data. For custom code that has been deposited into a community repository, provide accession details.* |
| Cell population abundance | *Describe the abundance of the relevant cell populations within post-sort fractions, providing details on the purity of the samples and how it was determined.* |
| Gating strategy | *Describe the gating strategy used for all relevant experiments, specifying the preliminary FSC/SSC gates of the starting cell population, indicating where boundaries between "positive" and "negative" staining cell populations are defined.* |

☐ Tick this box to confirm that a figure exemplifying the gating strategy is provided in the Supplementary Information.

# Magnetic resonance imaging

## Experimental design

| | |
|---|---|
| Design type | *Indicate task or resting state; event-related or block design.* |
| Design specifications | *Specify the number of blocks, trials or experimental units per session and/or subject, and specify the length of each trial or block (if trials are blocked) and interval between trials.* |
| Behavioral performance measures | *State number and/or type of variables recorded (e.g. correct button press, response time) and what statistics were used to establish that the subjects were performing the task as expected (e.g. mean, range, and/or standard deviation across subjects).* |

## Acquisition

| | |
|---|---|
| Imaging type(s) | *Specify: functional, structural, diffusion, perfusion.* |
| Field strength | *Specify in Tesla* |
| Sequence & imaging parameters | *Specify the pulse sequence type (gradient echo, spin echo, etc.), imaging type (EPI, spiral, etc.), field of view, matrix size, slice thickness, orientation and TE/TR/flip angle.* |
| Area of acquisition | *State whether a whole brain scan was used OR define the area of acquisition, describing how the region was determined.* |

Diffusion MRI   ☐ Used   ☐ Not used

## Preprocessing

| | |
|---|---|
| Preprocessing software | *Provide detail on software version and revision number and on specific parameters (model/functions, brain extraction, segmentation, smoothing kernel size, etc.).* |
| Normalization | *If data were normalized/standardized, describe the approach(es): specify linear or non-linear and define image types used for transformation OR indicate that data were not normalized and explain rationale for lack of normalization.* |
| Normalization template | *Describe the template used for normalization/transformation, specifying subject space or group standardized space (e.g. original Talairach, MNI305, ICBM152) OR indicate that the data were not normalized.* |
| Noise and artifact removal | *Describe your procedure(s) for artifact and structured noise removal, specifying motion parameters, tissue signals and physiological signals (heart rate, respiration).* |

| Volume censoring | *Define your software and/or method and criteria for volume censoring, and state the extent of such censoring.* |

## Statistical modeling & inference

| Model type and settings | *Specify type (mass univariate, multivariate, RSA, predictive, etc.) and describe essential details of the model at the first and second levels (e.g. fixed, random or mixed effects; drift or auto-correlation).* |

| Effect(s) tested | *Define precise effect in terms of the task or stimulus conditions instead of psychological concepts and indicate whether ANOVA or factorial designs were used.* |

Specify type of analysis: ☐ Whole brain    ☐ ROI-based    ☐ Both

| Statistic type for inference | *Specify voxel-wise or cluster-wise and report all relevant parameters for cluster-wise methods.* |

(See Eklund et al. 2016)

| Correction | *Describe the type of correction and how it is obtained for multiple comparisons (e.g. FWE, FDR, permutation or Monte Carlo).* |

## Models & analysis

| n/a | Involved in the study |
|---|---|
| ☐ | ☐ Functional and/or effective connectivity |
| ☐ | ☐ Graph analysis |
| ☐ | ☐ Multivariate modeling or predictive analysis |

| Functional and/or effective connectivity | *Report the measures of dependence used and the model details (e.g. Pearson correlation, partial correlation, mutual information).* |

| Graph analysis | *Report the dependent variable and connectivity measure, specifying weighted graph or binarized graph, subject- or group-level, and the global and/or node summaries used (e.g. clustering coefficient, efficiency, etc.).* |

| Multivariate modeling and predictive analysis | *Specify independent variables, features extraction and dimension reduction, model, training and evaluation metrics.* |

