## [Peer Review File · Nature Cell Biology]

Peer Review Information

Journal: Nature Cell Biology

Manuscript Title: LIANA+: an all-in-one cell-cell communication framework

Corresponding author name(s): Professor Julio Saez Rodriguez

Editorial Notes:

Transferred manuscripts This manuscript has been previously reviewed at another journal that is not operating a transparent peer review scheme. This document only contains reviewer comments, rebuttal and decision letters for versions considered at Nature Cell Biology.

Reviewer Comments & Decisions:

Feedback Before Review:

Dear Dr. Saez Rodriguez,

We now have received all reviews from the original reviewers on your revision "LIANA+: an all-in-one cell-cell communication framework" -- we have been discussing the reviewers' comments editorially and apologize for the delay in sending our decision to you.

I am writing because Reviewer #2 shared some persistent concerns that we found important regarding some of their original comments and how they were addressed in revision. I am pasting Rev#2's comments below.

At this stage, we are hoping to ask if you would be willing to please send us responses to the comments pasted below, as a point-by-point response to the reviewer's comments.

To be clear, we are NOT asking for a revised manuscript/new experiments/new data now. We are simply interested in hearing your thoughts on these points within 1-2 business days - including if you may have data that could address these points, or if you think you would be able to provide revisions along these lines in a reasonable time frame if needed.

This would be extremely informative to us as we continue the editorial process.

Please note that we will likely then discuss your response with the reviewer again before reaching our decision editorially.

Please let me know if you have any questions and I look forward to hearing your thoughts on the points below. Thank you so much for your time and consideration,

Best wishes,
Melina

--

Melina Casadio, PhD
Senior Editor, NCB

REV#2 re-review comments:

The authors provided better application instances to demonstrate the value of LIANA+ as a unified framework integrating multi-methods and multi-omics. The authors did refine the benchmark and scalability evaluation works, but these results cannot fully answer my concerns. Below are my specific comments.

Major comments:

1. Though we agree with the authors that "it is not possible with current benchmark data to definitely say which method performs best", there would still be some approaches to approximately compare the performances between LIANA+ with other individual methods, just like the indirect ground truth generation strategy mentioned in section 2.2. The purpose of integration in LIANA+ should be the improvement of result reliability but not the result recall rate, especially for the CCC predicting area. A further benchmark is still needed otherwise such integration could be ineffective. The same request is in the comparison with spatial CCC methods.

2. The authors should better demonstrate LIANA+'s scalability by displaying the total running time and memory usage of an entire workflow of LIANA+, including individual methods running and Bayesian multi-view factor analysis. Since one of the novelty of LIANA+ is integrating multi-omics data, its scalability while applying to data with different numbers of views should also be evaluated.

Minor comments:

1. Add a significant level to Fig 2E.
2. Scale the y-axis and adjust points to make Supplementary Figure S1 be more recognizable.
3. The boxes in Supplementary Figure S6 should be ranked to be more informative.

Author Response:

Legend:

Reviewer Comments

Response

REV#2 re-review comments:

The authors provided better application instances to demonstrate the value of LIANA+ as a unified framework integrating multi-methods and multi-omics. The authors did refine the benchmark and scalability evaluation works, but these results cannot fully answer my concerns. Below are my specific comments.

We thank the reviewer for recognizing the value of LIANA+ as an integrative framework and for the clarification of their previous comments.

Major comments:

1. Though we agree with the authors that “it is not possible with current benchmark data to definitely say which method performs best”, there would still be some approaches to approximately compare the performances between LIANA+ with other individual methods, just like the indirect ground truth generation strategy mentioned in section 2.2. The purpose of integration in LIANA+ should be the improvement of result reliability but not the result recall rate, especially for the CCC predicting area. A further benchmark is still needed otherwise such integration could be ineffective. The same request is in the comparison with spatial CCC methods.

As the reviewer says, the purpose of integration in LIANA+ is not about recall rate, or any other benchmark metric, but rather the reliability of the combination of methods. Since LIANA+ integrates diverse methods, it cannot be captured by a single statistic that can be benchmarked with the (>20) individual methods in LIANA+ itself. Instead, LIANA+ combines those complementary methods to enable novel analyses and improve the quality of the hypotheses generated by any individual method.

To better illustrate the integrative value of LIANA+, we will include a detailed evaluation (see example in **Supp. Figure X**) that demonstrates the significant reduction in prediction space achieved through each component’s (complimentary individual methods) unique contribution to the integrative analysis. Complex dataset applications, such as the one on single-cell and spatially-resolved data from myocardial infarction (section 2.2), best highlight the unique ability of LIANA+ to obtain a refined set of reliable interactions, consistent across methods and technologies.

Supplementary Figure X. LIANA+ application on Myocardial infarction substantially reduces the prediction space to yield an interpretable subset of ligand-receptors deregulated across the disease. **A) (1)** inference of the full possible prediction space, potentially reaching >5M ligand-receptor interactions across 110 cell-type pairs, 28 samples, and 3 conditions. **(2)** the incorporation of cell type pairs, observed to be spatially associated (*i*; *t*-value > 2; *R*² > 0.1), reduces the initial prediction space by ~5 fold. **(3)** Incorporating information about spatially-localized LR associated with disease (identified using **(ii)** local scores using NMF **(iii)**) further reduces the prediction space. **(4)** multi-view factorization on 10K LR pairs across ~20 cell types reduces prediction space down to ~500 interactions across ~5 cell type pairs, associated with disease progression. **(5)** Targeted differential expression analysis yields ~50 significantly deregulated LR in a specific pair of interacting cell types. **(6)** using a causal network search the prediction space is reduced further to the specific mechanistic hypothesis of 2 LR interactions associated with downstream signalling to ~5 fibrosis-related TFs. **B)** Bar plot showing the anticipated reduction in the total number of interactions until a set of reliable interactions is obtained (Step 5), and the subsequent translation of some of those into specific mechanistic hypotheses (Step 6).

Further, we apologize for not communicating clearly what we meant by “it is not possible with current benchmark data to definitely say which method performs best”, and for possibly being over-cautious or over-conservative with interpreting the results of our evaluations - which are to our knowledge the most comprehensive set of benchmarks in any cell-cell-communication manuscript.

Using four independent types of indirect ground truth (tissue pathology, cell type specificity, spatial co-localisation, and condition classification), we have already compared the performances of the single-cell and spatial components, as well as the individual methods, in LIANA+ (see **Supp. Figures 3, 6, and 7** below).

To address this point, we will adjust the discussion and results of our manuscript to better underline the results and insights from our benchmarks. Moreover, we could also specifically quantify the effectiveness of the integrative potential of LIANA+. In such case, we would

extend our current benchmarks to evaluate the degree to which the integration of the statistics provided by individual methods yields improved performance, and hence more reliable interactions.

In more detail, our benchmarks have shown that the methods unique to LIANA+ (e.g. spatially-weighted cosine similarity, rank aggregate, or intercellular multi-view factorization) exhibit comparable or better performance than the respective state-of-the-art methods. As such, we have explicitly compared the individual methods and components in LIANA, and thus support their effectiveness:

1) We have shown that the local metrics in LIANA+ (e.g. cosine similarity) perform on average better than e.g. SpatialDM (Moran's R) or scHOT (masked spearman) when predicting malignancy labels and cell type presence (**Supplementary Figure 3**).

Supplementary Figure S3. A) AUROC and B) weighted F1 when using local metrics to classify malignant spots in breast cancer data ⁹⁸; **C) R² and D) RMSE** when using local metrics to predict cell type proportions in heart data ⁴⁰. The line in the boxplots represents the median with hinges showing the first and third quartiles and the whiskers extend up to 1.5 times the interquartile range above and below the box hinges.

2) We have shown that all individual single-cell methods in LIANA+, including the rank aggregate proposed in our previous manuscript³, perform better than random. Using these benchmarks - which expand and improve previous benchmarks in the single-cell field³⁻⁵, we have underlined best practices and strategies to refine predictions into potentially more reliable insights from multi-modal and cross-technology data, enabled uniquely by the flexibility and modularity of LIANA+.

Supplementary Figure S6. Comparison of ligand-receptor inference methods using the spatial colocalisation of cell types and ligand-receptors as assumed truth. Each method and its individual scoring functions are represented by a different colour. **A)** Quantifies the performance of each methods' individual scoring functions using the Area Under the Receiver Operating Characteristic Curve (AUROC). **B)** Balanced Accuracy for each of the methods filtered according to their suggested false positive filtering thresholds. **C)** and **D)** measure normalised F1 score following filtering for each score (*) and method, respectively. For each metric, a score of 0.5, denoted by the dashed red line, indicates random performance. Note that while we show 0 to 1 for normalised F1, unlike Balanced Accuracy and AUROC, it is not bound to 1 (Methods). Both NATMI and Connectome use expression products (Product*) as a measure of magnitude strength. The central line in cyan within each box marks the median, with the box hinges representing the first and third quartiles. The whiskers extend up to 1.5 times the interquartile range above and below the box hinges. Outliers are depicted as individual diamond-shaped points beyond the whiskers.

3) We have shown that multi-view factorization has on average better classification performance than the current state-of-the-art⁶ across several individual methods and datasets. With the added benefit that it reports interactions per cell type pair (not a single set of importances for all cell types), and can be efficiently run on a standard laptop without GPU acceleration:

Supplementary Figure S7. A) Classification setup to evaluate the ability of ligand-receptor methods, combined with Tensor-cell2cell and MOFA+ to separate conditions from multi-condition atlases in an unsupervised manner. **B)** Area under the receiver-operator curve (AUROC) and **C)** weighted F1 score, as calculated across 5 datasets, and the 'Average' performance.

2. The authors should better demonstrate LIANA+'s scalability by displaying the total running time and memory usage of an entire workflow of LIANA+, including individual methods running and Bayesian multi-view factor analysis. Since one of the novelty of LIANA+ is integrating multi-omics data, its scalability while applying to data with different numbers of views should also be evaluated.

We can include a comprehensive benchmark detailing the anticipated total running time and memory usage for all standard workflows in LIANA+, including individual methods, Bayesian multi-view factor analysis, and multi-omics data with different numbers of views.

We performed a preliminary benchmark executed on a personal laptop from 2019, equipped with 16GB RAM and an INTEL i7 processor, see Table below. The maximum RAM usage reached 4GB, with the longest workflow running time being under 10 minutes. The multi-view factor analysis stands out for its computational efficiency (<3 minutes on a standard laptop), unlike competing algorithms that generally require GPU acceleration. Similarly, the integration of spatial multi-omics data was completed in just 3 minutes. We will expand this analysis.

Supplementary Table Y. Computational scalability of standard LIANA+ workflows.

Workflow	Data Details	Method Components	Total RAM usage†	Running Time‡
Steady-state single-cell data	5k PBMCs	LIANA single-cell Rank aggregate* of 5 methods with 1,000 permutations, visualization	2GB	3 minutes
Identification of bivariate spatial co-expression patterns	A myocardial infarction 10X Visium slide	Estimation of global and local co-localisation statistics, corresponding categories and permutation p-values. Subsequent, decomposition of local scores into spatial CCC patterns using NMF. Pathway enrichment and co-localisation with TFs inferred to be active.	4GB	9 minutes
Learning spatial relationships across views	A myocardial infarction 10X Visium slide	Exploratory analysis leveraging linear and random forest models for diverse modeling designs.	4GB	5 minutes
Learning spatial relationships and identifying local bivariate associations from multi-omics data	Joint 10X Visium and MALDI-MSI data (~3k spots each)	Multi-view learning, bivariate spatial associations	4GB	3 minutes
Multi-view factorization of CCC patterns	16 samples from Lupus patients, with a total of 15k cells	Rank aggregate* of 5 single-cell methods, multi-view factorisation, Pathway enrichment	3GB	3 minutes
Decomposition of CCC patterns with Tensor-cell2cell	16 samples from Lupus patients, with a total of 15k cells	Rank aggregate* of 5 single-cell methods, PARAFAC factorisation, Pathway enrichment	3GB	4 minutes (with GPU)
Differential expression analysis, with causal network inference	16 samples from Lupus patients, with a total of 15k cells	Identification of DE ligand-receptor interactions, inference of putative causal connections to downstream TFs	4GB	<10 minutes

* LIANA's rank aggregate, published in our previous work, can be replaced by any of the other 8 methods in LIANA+, including CellPhoneDB, CellChat, or scSeqComm.
† Reported statistics were approximated and may change once we do a formal run.

Minor comments:

1. Add a significant level to Fig 2E.

We will perform a t-test and illustrate the significance level in the figure.

2. Scale the y-axis and adjust points to make Supplementary Figure S1 be more recognizable.

We will adjust the y-axis such that the points are visible.

3. The boxes in Supplementary Figure S6 should be ranked to be more informative.

We will rank the boxes in Supp. Fig. S6 according to their median performance.

Bibliography

1. Wu, S. Z. *et al.* A single-cell and spatially resolved atlas of human breast cancers. *Nature Genetics* (2021).
2. Kuppe, C. *et al.* Spatial multi-omic map of human myocardial infarction. *Nature* **608**, 766–777 (2022).
3. Dimitrov, D. *et al.* Comparison of methods and resources for cell-cell communication inference from single-cell RNA-Seq data. *Nat. Commun.* **13**, 3224 (2022).
4. Liu, Z., Sun, D. & Wang, C. Evaluation of cell-cell interaction methods by integrating single-cell RNA sequencing data with spatial information. *Genome Biol.* **23**, 218 (2022).
5. Luo, J., Deng, M., Zhang, X. & Sun, X. ESICCC as a systematic computational framework for evaluation, selection, and integration of cell-cell communication inference methods. *Genome Res.* (2023) doi:10.1101/gr.278001.123.
6. Armingol, E. *et al.* Context-aware deconvolution of cell-cell communication with Tensor-cell2cell. *Nat. Commun.* **13**, 3665 (2022).

Decision Letter, initial version:

Dear Professor Saez Rodriguez,

I am writing on behalf of my colleague Dr Melina Casadio, who is currently out of the office. Your manuscript, "LIANA+: an all-in-one cell-cell communication framework", has now been seen by our original referees, who are experts in genomics (referee 1); and bioinformatics (referee 2). As you will see from their comments (attached below) they find this work of interest, but have raised some important points. Although we are also very interested in this study, we believe that their concerns should be addressed before we can consider publication in Nature Cell Biology.

Nature Cell Biology editors discuss the referee reports in detail within the editorial team, including the chief editor, to identify key referee points that should be addressed with priority, and requests that are overruled as being beyond the scope of the current study. To guide the scope of the revisions, I have listed these points below. We are committed to providing a fair and constructive peer-review process, so please feel free to contact me if you would like to discuss any of the referee comments further.

All other referee concerns pertaining to strengthening existing data, methodological details, clarifications and textual changes, should also be addressed.

Finally please pay close attention to our guidelines on statistical and methodological reporting (listed below) as failure to do so may delay the reconsideration of the revised manuscript. In particular please provide:

- a Supplementary Figure including unprocessed images of all gels/blots in the form of a multi-page pdf file. Please ensure that blots/gels are labeled and the sections presented in the figures are clearly indicated.
- a Supplementary Table including all numerical source data in Excel format, with data for different figures provided as different sheets within a single Excel file. The file should include source data giving rise to graphical representations and statistical descriptions in the paper and for all instances where the figures present representative experiments of multiple independent repeats, the source data of all repeats should be provided.

We therefore invite you to take these points into account when revising the manuscript. In addition, when preparing the revision please:

- ensure that it conforms to our format instructions and publication policies (see below and <https://www.nature.com/nature/for-authors>).
- provide a point-by-point rebuttal to the full referee reports verbatim, as provided at the end of this letter.
- provide the completed Reporting Summary (found here <https://www.nature.com/documents/nr-reporting-summary.pdf>). This is essential for reconsideration of the manuscript and will be available to editors and referees in the event of peer review. For more information

see <http://www.nature.com/authors/policies/availability.html> or contact me.

When submitting the revised version of your manuscript, please pay close attention to our [href="https://www.nature.com/nature-portfolio/editorial-policies/image-integrity">Digital Image Integrity Guidelines](https://www.nature.com/nature-portfolio/editorial-policies/image-integrity). and to the following points below:

Nature Cell Biology is committed to improving transparency in authorship. As part of our efforts in this direction, we are now requesting that all authors identified as 'corresponding author' on published papers create and link their Open Researcher and Contributor Identifier (ORCID) with their account on the Manuscript Tracking System (MTS), prior to acceptance. ORCID helps the scientific community achieve unambiguous attribution of all scholarly contributions. You can create and link your ORCID from the home page of the MTS by clicking on 'Modify my Springer Nature account'. For more information please visit www.springernature.com/orcid.

This journal strongly supports public availability of data. Please place the data used in your paper into a public data repository, or alternatively, present the data as Supplementary Information. If data can only be shared on request, please explain why in your Data Availability Statement, and also in the correspondence with your editor. Please note that for some data types, deposition in a public repository is mandatory - more information on our data deposition policies and available repositories appears below.

[Redacted]

We would like to receive the revision within four weeks. If submitted within this time period, reconsideration of the revised manuscript will not be affected by related studies published elsewhere, or accepted for publication in Nature Cell Biology in the meantime. We would be happy to consider a revision even after this timeframe, but in that case we will consider the published literature at the time of resubmission when assessing the file.

We hope that you will find our referees' comments, and editorial guidance helpful. Please do not hesitate to contact me if there is anything you would like to discuss.

Best wishes,

Sabrya Carim

Sabrya Carim, PhD
(she/her/hers)
Associate Editor, Nature Cell Biology
Nature Portfolio

Springer Nature
The Campus, 4 Crinan Street, London N1 9XW, UK
sabrya.carim@springernature.com
<https://orcid.org/0000-0001-9485-1938>

Reviewers' Comments:

Reviewer #1:

Remarks to the Author:

The writing and figure legends have greatly improved.

Comments:

Figure 2.

1. For the reader please mention which spatial technologies were used (Visium and MALDI-MSI).

2. Fig 2D - why is there no violin for lesioned

3. Deconvolution must have been used in Fig 2 but there is no mention of how it was done.

4. What are the dimensions of the data? how many metabolites? How many receptors? How many cell types? How are cell type proportions inferred from the spatial data?

5. The example focuses on Dopamine and mentions other metabolites but this is not provided as a supplementary data. How many metabolite - receptor - cell type predictions did LIANA+ identify and is dopamine at the top? or is it focused on because the original experiment specifically targeted dopamine producing neurons?

6. About the predictions that dopamine levels are predicted by MSN1 and MSN2 cell abundance and Drd2 expression. Do MSN1 and MSN2 express Drd2? Do they produce dopamine? What confidence is there in the deconvoluted MSN1 and MSN2 cell type proportions (estimated from de-convolution of Visium data?). Is there any danger that there is remaining signal from dopaminergic neurons that can explain the prediction?

7. In addition to G, H and I it would be useful to show the underlying spatial transcriptomics data for the receptors in question and the spatial metabolomics data for dopamine. It should also state explicitly which receptors and which metabolites are being examined (perhaps label C?).

Figure 3.

8. Supplementary data with the full set of predictions should be provided.

9. Fig3L shows 'Deregulated interactions of interest' but it is unclear whether this is only a subset of predicted pairs and whether these were more highly ranked than others.

From the previous review.

10. It is still critical that the ligand receptor pairs and their sources (pubmed or DOIs) are provided as a supplementary data file with this publication. This is needed for reproducibility. The authors have suggested the data is easy to extract from the tool however the version used in this publication will likely not persist as the database is updated. There is also no guarantee that the knowledgebase used for this publication will remain open to the public in the future.

Making the data available ensures the reproducibility of the research and that the level of evidence for the pairs is clear. It also ensures that the authors of prior annotation efforts are acknowledged.

Minor:

"We noted similar contributions for per view the remainder of the well-explained metabolite peaks " Not sure what 'for per view means'.

Reviewer #2:

Remarks to the Author:

The authors provided better application instances to demonstrate the value of LIANA+ as a unified framework integrating multi-methods and multi-omics. The authors did refine the benchmark and scalability evaluation works, but these results cannot fully answer my concerns. Below are my specific comments.

Major comments:

1. Though we agree with the authors that "it is not possible with current benchmark data to definitely say which method performs best", there would still be some approaches to approximately compare the performances between LIANA+ with other individual methods, just like the indirect ground truth generation strategy mentioned in section 2.2. The purpose of integration in LIANA+ should be the improvement of result reliability but not the result recall rate, especially for the CCC predicting area. A further benchmark is still needed otherwise such integration could be ineffective. The same request is in the comparison with spatial CCC methods.

2. The authors should better demonstrate LIANA+'s scalability by displaying the total running time and memory usage of an entire workflow of LIANA+, including individual methods running and Bayesian multi-view factor analysis. Since one of the novelty of LIANA+ is integrating multi-omics data, its scalability while applying to data with different numbers of views should also be evaluated.

Minor comments:

1. Add a significant level to Fig 2E.

2. Scale the y-axis and adjust points to make Supplementary Figure S1 be more recognizable.

3. The boxes in Supplementary Figure S6 should be ranked to be more informative.

**Additional comments from reviewer#2 after assessing revision plan:

I have now had the opportunity to thoroughly review the authors' proposed revision plan.

Upon reviewing the attached file containing the authors' responses to my previous comments and concerns, I am pleased to note that the authors have presented a strategic approach for generating a reliable interaction set, which would allow for a comprehensive evaluation of LIANA+. If the proposed analyses are implemented as outlined by the authors, then my concerns about the integration benchmark can probably be addressed. Regarding scalability, the authors provided a detailed summary, and I am surprised by the high computational efficiency of multi-view factor analysis. If the approximate running time remains relatively unchanged during the formal run, then I am confident that my concerns in this area will also be adequately addressed.

In summary, I find the authors' proposal to be compelling and capable of addressing the issues I raised during my re-review. I believe these suggested changes would constitute a minor revision that significantly enhances the quality of the manuscript.

GUIDELINES FOR SUBMISSION OF NATURE CELL BIOLOGY ARTICLES

ARTICLE FORMAT

ABSTRACT – should not exceed 150 words and should be unreferenced. This paragraph is the most visible part of the paper and should briefly outline the background and rationale for the work, and accurately summarize the main results and conclusions. Key genes, proteins and organisms should be specified to ensure discoverability of the paper in online searches.

TEXT – the main text consists of the Introduction, Results, and Discussion sections and must not exceed 3500 words including the abstract. The Introduction should expand on the background relating to the work. The Results should be divided in subsections with subheadings, and should provide a concise and accurate description of the experimental findings. The Discussion should

expand on the findings and their implications. All relevant primary literature should be cited, in particular when discussing the background and specific findings.

REFERENCES – are limited to a total of 70 in the main text and Methods combined,. They must be numbered sequentially as they appear in the main text, tables and figure legends and Methods and must follow the precise style of Nature Cell Biology references. References only cited in the Methods should be numbered consecutively following the last reference cited in the main text. References only associated with Supplementary Information (e.g. in supplementary legends) do not count toward the total reference limit and do not need to be cited in numerical continuity with references in the main text. Only published papers can be cited, and each publication cited should be included in the numbered reference list, which should include the manuscript titles. Footnotes are not permitted.

Methods should be written concisely, but should contain all elements necessary to allow interpretation and replication of the results. As a guideline, Methods sections typically do not exceed 3,000 words. The Methods should be divided into subsections listing reagents and techniques. When citing previous methods, accurate references should be provided and any alterations should be noted. Information must be provided about: antibody dilutions, company names, catalogue numbers and clone numbers for monoclonal antibodies; sequences of RNAi and cDNA probes/primers or company names and catalogue numbers if reagents are commercial; cell line names, sources and information on cell line identity and authentication. Animal studies and experiments involving human subjects must be reported in detail, identifying the committees approving the protocols. For studies involving human subjects/samples, a statement must be included confirming that informed consent was obtained. Statistical analyses and information on the reproducibility of experimental results should be provided in a section titled "Statistics and Reproducibility".

All Nature Cell Biology manuscripts submitted on or after March 21 2016, must include a Data availability statement as a separate section after Methods but before references, under the heading "Data Availability". For Springer Nature policies on data availability see <http://www.nature.com/authors/policies/availability.html>; for more information on this particular policy see <http://www.nature.com/authors/policies/data/data-availability-statements-data-citations.pdf>. The Data availability statement should include:

- Accession codes for primary datasets (generated during the study under consideration and designated as "primary accessions") and secondary datasets (published datasets reanalysed during the study under consideration, designated as "referenced accessions"). For primary accessions data should be made public to coincide with publication of the manuscript. A list of data types for which submission to community-endorsed public repositories is mandated (including sequence, structure, microarray, deep sequencing data) can be found here <http://www.nature.com/authors/policies/availability.html#data>.
- Unique identifiers (accession codes, DOIs or other unique persistent identifier) and hyperlinks for datasets deposited in an approved repository, but for which data deposition is not mandated (see here for details <http://www.nature.com/sdata/data-policies/repositories>).
- At a minimum, please include a statement confirming that all relevant data are available from the authors, and/or are included with the manuscript (e.g. as source data or supplementary information), listing which data are included (e.g. by figure panels and data types) and mentioning any restrictions on availability.
- If a dataset has a Digital Object Identifier (DOI) as its unique identifier, we strongly encourage including this in the Reference list and citing the dataset in the Methods.

We recommend that you upload the step-by-step protocols used in this manuscript to the Protocol Exchange. More details can be found at www.nature.com/protocolexchange/about.

DISPLAY ITEMS – main display items are limited to 6-8 main figures and/or main tables. For Supplementary Information see below.

FIGURES – Colour figure publication costs \$395 per colour figure. All panels of a multi-panel figure must be logically connected and arranged as they would appear in the final version. Unnecessary figures and figure panels should be avoided (e.g. data presented in small tables could be stated briefly in the text instead).

All imaging data should be accompanied by scale bars, which should be defined in the legend. Cropped images of gels/blots are acceptable, but need to be accompanied by size markers, and to retain visible background signal within the linear range (i.e. should not be saturated). The boundaries of panels with low background have to be demarked with black lines. Splicing of panels should only be considered if unavoidable, and must be clearly marked on the figure, and noted in the legend with a statement on whether the samples were obtained and processed simultaneously. Quantitative comparisons between samples on different gels/blots are discouraged; if this is unavoidable, it has to be performed for samples derived from the same experiment with gels/blots were processed in parallel, which needs to be stated in the legend.

- For line art, graphs, charts and schematics we prefer Adobe Illustrator (.AI), Encapsulated PostScript (.EPS) or Portable Document Format (.PDF). Files should be saved or exported as such directly from the application in which they were made, to allow us to restyle them according to our journal house style.
- We accept PowerPoint (.PPT) files if they are fully editable. However, please refrain from adding PowerPoint graphical effects to objects, as this results in them outputting poor quality raster art. Text used for PowerPoint figures should be Helvetica (preferred) or Arial.
- We do not recommend using Adobe Photoshop for designing figures, but we can accept Photoshop generated (.PSD or .TIFF) files only if each element included in the figure (text, labels, pictures, graphs, arrows and scale bars) are on separate layers. All text should be editable in 'type layers' and line-art such as graphs and other simple schematics should be preserved and embedded within 'vector smart objects' - not flattened raster/bitmap graphics.
- Some programs can generate Postscript by 'printing to file' (found in the Print dialogue). If using an application not listed above, save the file in PostScript format or email our Art Editor, Allen Beattie for advice (a.beattie@nature.com).

Regardless of format, all figures must be vector graphic compatible files, not supplied in a flattened raster/bitmap graphics format, but should be fully editable, allowing us to highlight/copy/paste all text and move individual parts of the figures (i.e. arrows, lines, x and y axes, graphs, tick marks, scale bars etc). The only parts of the figure that should be in pixel raster/bitmap format are photographic images or 3D rendered graphics/complex technical illustrations.

Unprocessed scans of all key data generated through electrophoretic separation techniques need to be presented in a supplementary figure that should be labeled and numbered as the final supplementary figure, and should be mentioned in every relevant figure legend. This figure does not count towards the total number of figures and is the only figure that can be displayed over multiple pages, but should be provided as a single file, in PDF or TIFF format. Data in this figure can be displayed in a relatively informal style, but size markers and the figures panels corresponding to the presented data must be indicated.

The total number of Supplementary Figures (not including the “unprocessed scans” Supplementary Figure) should not exceed the number of main display items (figures and/or tables (see our Guide to Authors and March 2012 editorial <http://www.nature.com/ncb/authors/submit/index.html#suppinfo>; <http://www.nature.com/ncb/journal/v14/n3/index.html#ed>). No restrictions apply to Supplementary Tables or Videos, but we advise authors to be selective in including supplemental data.

GUIDELINES FOR EXPERIMENTAL AND STATISTICAL REPORTING

REPORTING REQUIREMENTS – We ask authors to complete a Reporting Summary that collects information on experimental design and reagents. We hope this will aid in your evaluation of the paper. The Reporting Summary can be found here <https://www.nature.com/documents/nr-reporting-summary.pdf>) Please note that these forms are dynamic ‘smart pdfs’ and must therefore be downloaded and completed in Adobe Reader. We will then flatten them for ease of use. If you would like to reference the guidance text as you complete the template, please access these flattened versions at <http://www.nature.com/authors/policies/availability.html>.

Print Email

Author Rebuttal to Initial comments

Legend:

Green - Reviewer Comments

Black - Authors' Response

Blue - Manuscript Text Changes

Reviewers' Comments:

Reviewer #1:

Remarks to the Author:

The writing and figure legends have greatly improved.

We thank the reviewer for acknowledging the improvements in our manuscript.

Comments:

Figure 2.

1. For the reader please mention which spatial technologies were used (Visium and MALDI-MSI).

We have updated the main text to reflect the technologies used:

“ [...] We illustrate the joint application of multi-view modelling and local metrics to study metabolite-mediated interactions from multi-omics data using a recent murine Parkinson's disease model dataset ¹. This spatially-resolved dataset provides metabolome and transcriptome measurements, respectively generated using MALDI mass spectrometry imaging and 10X Visium technologies (Fig. 2A). [...] “

As well as the figure legend:

*[...] **Figure 2. LIANA+ models intercellular communication using spatial multi-omics data. A)** Spatially-resolved transcriptomics (10X Visium) and metabolomics (MALDI mass spectrometry imaging) captured on the same tissue section yield two matrices, the observations of which are in different tissue locations ¹. [...]*

2. Fig2D - why is there no violin for lesioned

Thanks for pointing this out. The violin plot for lesioned was not visible because of the very low R^2 values. To address this, we have adjusted the y-axis and also replaced the violins with boxplots so that the points and distribution for the lesioned hemisphere are visible. Note that we have swapped **Fig2D** with plots to show the spatial distribution of Dopamine and Drd2 (see point 7); the data that was before in Fig2D is now within **Supp. Fig S4A**, where we highlight it in dark red.

Supplementary Figure S4. Multi-view modelling prediction performance per target in spatially-resolved metabolite-transcriptome data. **A)** Top 10 metabolite peaks with the highest variance explained (R^2), including Dopamine (in dark red) and 3-methoxytyramine (3-MT) peaks, validated in the original publication ¹. [...] The central line within each box marks the median, with the box hinges representing the first and third quartiles. The whiskers extend up to 1.5 times the interquartile range above and below the box hinges.

3. Deconvolution must have been used in Fig 2 but there is no mention of how it was done.

We apologise for the lack of clarity. We have now also added an explanation in the main text:

“[...] Using multi-view modelling, we inferred spatial relationships between metabolites, their corresponding brain-specific metabolite receptors, and cell types across different spatial contexts (views) (Fig. 2C; Methods). Specifically, we trained a model that predicts metabolite intensities using spatially-adjacent receptor expression and cell type proportions, deconvoluted using Tangram ² with a murine brain single-cell atlas ³ as a reference. [...]”

We also included analogous information in the figure legend:

“ [...] **C)** Multi-view modelling integrates metabolite peak intensities, brain-specific receptor expression, and cell type proportions to identify spatial relationships. This approach enables the estimation of joint performance and individual contributions of receptor expression and cell-type proportions in predicting metabolite peak intensities. Cell-type proportions were deconvoluted ² using a murine single-cell atlas ³ as a reference. [...]”

This information is expanded in the already existing description of the deconvolution approach in the Methods.

4. What are the dimensions of the data? how many metabolites? How many receptors?

This was indeed not sufficiently detailed in the manuscript. The metabolome replicates contain 2,483-3,040 spots each with ~1,200 untargeted metabolite peaks (m/z spectra were rounded to the third decimal). Out of these peaks, several were validated and annotated as specific metabolites, including Dopamine and its metabolite 3-methoxytyramine ¹. In our multi-view model, we predicted the 250 most variable metabolite peaks that were shared across the slides, which resulted in 83 shared highly-variable peaks.

The transcriptome replicates contain 2,577-3,036 spots each with ~16,500 genes after filtering. Out of these genes, 45 were metabolite receptors located in the brain with some variance in all slides.

We have now extended the relevant section in the Methods:

“ [...] We focused on the intersection of the top 250 highly-variable metabolite peaks (targets) across the three slides and excluded any predictors with little-to-no variation - i.e. genes not within the top 12500 highly-variable genes; and cell types with a coefficient of variation below the 20th percentile. Brain-specific receptors were obtained from MetalinksDB ⁴, customised to include only metabolites found in the brain or cerebrospinal fluid. After the preprocessing steps, using our multi-view modeling procedure, we analysed 83 metabolite peaks as targets, along with 45 receptors and 37 cell types as predictors. [...]”

How many cell types? How are cell type proportions inferred from the spatial data?

We used a single-cell murine brain atlas with 48 refined cell type annotations from Zeisel et al ³ to deconvolute the cell types per spot in the 10X Visium data. Out of these, for the multi-view modelling, we kept only 35 that showed at least some variance across the spots (see above).

We have now extended the relevant Methods section to include the details about the single-cell reference used for deconvolution:

“[...] We inferred cell type proportions using Tangram’s cell cluster level approach ², fit with 1,000 epochs and a learning rate of 0.1. As a reference for deconvolution, we used an annotated single-cell dataset from Zeisel et al ³. Specifically, we used the “TaxonomyRank4” cell group label, along with subgroups for dopaminergic and medium spiny neurons, which resulted in 48 refined murine brain cell types. [...]”

5. The example focuses on Dopamine and mentions other metabolites but this is not provided as a supplementary data.

We have now added another two tabs ‘SuppDataFig2_interactions’ and ‘SuppDataFig2_metrics’ which include the importance statistics and performance metrics for all metabolites and their predictors, respectively.

How many metabolite - receptor - cell type predictions did LIANA+ identify and is dopamine at the top? or is it focused on because the original experiment specifically targeted dopamine producing neurons?

LIANA+ identified ~40,000 metabolite-receptor-cell-type predictions across 83 metabolites and three slides. Among those 83 metabolites, LIANA+ identified the validated Dopamine peak as the fourth best-explained metabolite peak (see **Supp. Figure 4A** above), following 3-Methoxytyramine (a metabolite derived from Dopamine) and two unannotated m/z peaks. One of the two unannotated peaks (m/z 674.29) has a similar m/z ratio as the validated Dopamine peak (674.28), and is potentially also a peak that corresponds to dopamine.

As such, Dopamine was indeed the target of interest in the original experiment, and LIANA+ accurately identified it as relevant. We have now revised the relevant section's conclusion to reflect these points:

"[...] In conclusion, using LIANA+ we captured perturbation-driven changes in Dopamine's distribution and its associations with its canonical Drd2 receptor and MSN cell types 1 and 2. We also pinpointed the specific regions where these interactions take place, recapitulating and extending perturbed dopamine-signalling hypotheses ¹, and illustrating how LIANA+ enables novel analyses in spatial multi-omics data. [...]"

6. About the predictions that dopamine levels are predicted by MSN1 and MSN2 cell abundance and Drd2 expression. Do MSN1 and MSN2 express Drd2? Do they produce dopamine?

Indeed, Medium spiny neurons (MSNs) 1 and 2 predominantly express dopamine receptors Drd1 and Drd2, which were respectively used to classify MSN subtypes ³. While we focused on Drd2 and MSNs 1/2 as they were the top three predictors of dopamine, we also found Drd1 to be positively associated with Dopamine (median t-value=1.9, rank=7), which is coherent with it being a marker of MSN1 ³.

Dopamine is primarily produced by dopaminergic neurons located in the ventral tegmental area and the substantia nigra, which project to the Striatum - the area of focus in **Figure 2** and the largest component of the basal ganglia. Thus, the MSNs in the Striatum are not dopamine producers; instead, they are predominantly GABAergic neurons, which are receivers of Dopamine signaling involved in motor and cognitive functions, and hence important in Parkinson's disease ⁵.

We have updated the text to reflect these points:

"[...] Specifically, in the intact striatum, we found that the three best predictors of Dopamine (Median t-value ≥ 3) were dorsal medium-sized spiny neurons (MSN) 1 and 2 ³, and the Drd2 dopamine receptor (**Fig. 2F**). This reflects anticipated associations with Dopamine, as Drd2 is a canonical receptor of Dopamine. Similarly, GABAergic MSNs 1/2 are key receivers of dopamine signalling and were classified as types D1 or D2 according to the dopamine receptor (Drd1 or Drd2, respectively) they express ³. [...]"

What confidence is there in the deconvoluted MSN1 and MSN2 cell type proportions (estimated from de-convolution of Visium data?).

We deconvoluted the cell types with Tangram, which we chose due to its efficiency, good performance in benchmarks ⁶, and the heavy focus on murine brain data in its publication ². Moreover, in the original publication of the metabolome-transcriptome data ¹, the authors used Stereoscope - another deconvolution method ⁷ and reported similar deconvolution results to ours. As such, while deconvoluted cell type proportions remain predictions, our results are robust across different tools.

Is there any danger that there is remaining signal from dopaminergic neurons that can explain the prediction?

This is a good question and to address it we checked for dopaminergic neuron distributions in the murine brain slides. We saw that they were not spatially associated with Dopamine (**Rebuttal Figure 1**), which is anticipated as one would expect mostly cholinergic and GABAergic neurons in the striatum. This was also confirmed by our multi-view model as dopaminergic neuron proportions were generally weak predictors of dopamine with t-values < 1. As such, our results suggest that dopaminergic neurons are unlikely predictors of dopamine in the striatum. Furthermore, the binding of Dopamine to dopamine receptors of MSNs is a well-documented biological mechanism.

Rebuttal Figure 1. Spatial Distributions of dopamine and dopaminergic neurons in the striatum. Deconvolution results for Midbrain dopaminergic neurons (MBDOP) 1/2, olfactory bulb Dopaminergic periglomerular interneuron (OBDOP) 1/2, Cholinergic and monoaminergic neurons (CAMN), along with Dopamine intensity proportions are shown across all three slides included in the analysis presented in **Figure 2**.

7. In addition to G, H and I it would be useful to show the underlying spatial transcriptomics data for the receptors in question and the spatial metabolomics data for dopamine. It should also state explicitly which receptors and which metabolites are being examined (perhaps label C?).

We agree that such clarification would improve the quality of **Figure 2**. Therefore, we have now added labels to explicitly show that the interaction between Dopamine and Drd2 is presented in subpanels G, H, and I, along with examples for each of the variables represented in label C.

We have also moved Dopamine intensities and Drd2 expression from **Supp. Figure 5** to **Figure 2** (shown below); subsequently, we have added the spatial distributions of MSN1 and MSN2 to **Supp. Figure 5**.

Figure 2. *LIANA+ models intercellular communication using spatial multi-omics data.* **A)** Spatially-resolved transcriptomics (10X Visium) and metabolomics (MALDI mass spectrometry imaging) captured on the same tissue section yield two matrices, the observations of which are in different tissue locations ¹. **B)** Parkinson's disease mouse model annotated for striatum in intact and lesioned hemispheres. **C)** Multi-view modelling integrates metabolite peak intensities, brain-specific receptor expression, and cell type proportions to identify spatial relationships. This approach enables the estimation of joint performance and individual contributions of receptor expression and cell-type proportions in predicting metabolite peak intensities. Cell-type proportions were deconvoluted ² using a murine single-cell atlas ³ as a reference. **D)** Dopamine predictors ranked according to their median importance (y-axis; ordinary least squares t-values), with names shown for the top three predictors: *Drd2*, and *Medium Spiny Neurons (MSN) 1 and 2*. **E)** Normalised Dopamine peak intensities. **F)** Log_{1p} *Drd2* receptor expression. **G-I)** Local interactions between Dopamine and its canonical *Drd2* receptor as measured by spatially-weighted cosine similarity (**G**), its corresponding uncorrected permutation P-values (**H**), and categories (**I**). Figure panels showcase slide B1 from experiment V11L12-109 ³.

Figure 3.

8. Supplementary data with the full set of predictions should be provided.

We have now added the following tabs to the supplementary data excel sheet:

- SuppDataFig3_NMFscores - factor scores for the non-negative matrix factorisation (NMF) on local LIANA+ scores
- SuppDataFig3_NMFloadings - factor loadings for the NMF on local LIANA+ scores
- SuppDataFig3_LRs - ligand-receptor predictions across all single-nuc samples
- SuppDataFig3_MVloadings - all factor loadings from the multi-view factorisation, while all factor scores are in tab Fig3I
- SuppDataFig3_deaLRs - full ligand-receptor DE analysis interaction results, which were consistent with the highly ranked interactions from spatial NMF analysis (within the 95th percentile in any of the factors)

Moreover, we provide links to all public datasets, conda environment recipes, and all analysis code to further enable the reproducibility of our work (https://github.com/saezlab/lianaplus_manuscript).

9. Fig3L shows 'Deregulated interactions of interest' but it is unclear whether this is only a subset of predicted pairs and whether these were more highly ranked than others.

While all interactions reported in **Fig3L** showed a significant change in the expression of their ligands, receptors, or both across the conditions (FDR < 0.05), we focused on them because they were also highlighted by preceding analysis steps, specifically being highly ranked in the spatial data analysis, and their ligands/receptors were previously reported to play a role in fibrosis ⁸⁻¹⁰.

We have now clarified this in the figure legend:

“[...] **L)** *A subset of interactions, the ligand and/or receptors of which are known to play a role in fibrosis ⁸⁻¹⁰ and were significantly deregulated in fibroblast and/or myeloid cell types. Only interactions with the highest loadings (within the top 95th percentile) from the untargeted NMF analysis on spatially-informed local ligand-receptor interactions were included in the differential expression analysis. [...]*”

From the previous review.

10. It is still critical that the ligand receptor pairs and their sources (pubmed or DOIs) are provided as a supplementary data file with this publication. This is needed for reproducibility. The authors have suggested the data is easy to extract from the tool however the version used in this publication will likely not persist as the database is updated. There is also no guarantee that the knowledgebase used for this publication will remain open to the public in the future.

Making the data available ensures the reproducibility of the research and that the level of evidence for the pairs is clear. It also ensures that the authors of prior annotation efforts are acknowledged.

We agree with the reviewer that ensuring reproducibility and acknowledging the efforts of the previous authors is essential. As such, we have now included in the Supplementary Data Table the 'LigandReceptorPK' and 'MetaboliteReceptorPK' tabs which respectively contain the protein-mediated ligand-receptor and metabolite-receptor prior knowledge used in this work, along with the original database sources and pubmed IDs for literature-curated interactions. We also included the 'TFtargetPK' and 'ProteinProteinPK' for transcription factor and protein-protein interaction prior knowledge, respectively.

Minor:

"We noted similar contributions for per view the remainder of the well-explained metabolite peaks "

Not sure what 'for per view means'.

Apologies for the oversight. We now have clarified that '*for per view means*' refers to the contributions of different cell types and receptors (as predictors) of the metabolites:

"[...] We noted that receptors and cell types had similar prediction contributions for the remainder of the well-explained metabolite peaks (**Supp. Fig. S4B**). [...]"

Reviewer #2:

Remarks to the Author:

The authors provided better application instances to demonstrate the value of LIANA+ as a unified framework integrating multi-methods and multi-omics. The authors did refine the benchmark and scalability evaluation works, but these results cannot fully answer my concerns. Below are my specific comments.

We thank the reviewer for recognizing the value of LIANA+ as an integrative framework and for the additional feedback.

Major comments:

1. Though we agree with the authors that “it is not possible with current benchmark data to definitely say which method performs best”, there would still be some approaches to approximately compare the performances between LIANA+ with other individual methods, just like the indirect ground truth generation strategy mentioned in section 2.2. The purpose of integration in LIANA+ should be the improvement of result reliability but not the result recall rate, especially for the CCC predicting area. A further benchmark is still needed otherwise such integration could be ineffective. The same request is in the comparison with spatial CCC methods.

As the reviewer says, the purpose of integration in LIANA+ should not be about recall rate, or any other benchmark metric, but rather the reliability improvement when combining distinct LIANA+ components (complimentary individual methods). We developed LIANA+ with this intent in mind. Since LIANA+ integrates diverse methods, its performance cannot be captured by a single statistic that can then be benchmarked with the (>20) individual methods in LIANA+ itself. Instead, LIANA+ combines those complementary methods to enable novel analyses and improve the quality of the hypotheses generated by any individual method.

To address the reviewer’s comment, we have included a detailed evaluation that demonstrates the substantial reduction in prediction space achieved through each component’s unique contribution to the integrative analysis (**Supp. Figure S10A; Supp. Note 5**), therefore narrowing the results to the most reliable ones.

We have also quantified the integrative effectiveness of LIANA+ using an unsupervised clustering task of myeloid samples from heart failure patients versus healthy controls across multiple heart failure atlases (**Supp. Figure S10B-D**).

Our evaluation showed a clear increase in the Silhouette scores over individual methods (Global SpatialDM and CellPhoneDB) when we integrated different single-cell and spatial LIANA+ components (Step 3), with the median performance across datasets significantly exceeding a random baseline (z-score > 1.645; **Supp. Figure S10B**). In step 6, which is the last step of our analysis, combining the ligand-receptor and cell-type predictions from single-cell and spatial data with downstream signaling events, we saw the highest Adjusted Rand Index (ARI) across all steps (**Supp. Figure S10C**). Here, the integrative predictions from LIANA+ did better than random in all six datasets, with four of those being close to or significantly better (< 0.05) than the random predictions (**Supp. Figure S10D**).

This evaluation highlights the unique integrative potential of LIANA+ and enhanced reliability when working with complex dataset applications, such as the one on single-cell and spatially-resolved data from myocardial infarction (Section 2.2).

Supplementary Figure S10. LIANA+ generates interpretable and reliable predictions from complex datasets. A) Applying LIANA+ on Myocardial infarction substantially reduced the prediction space to yield an interpretable subset of predictions. (1) inference of the full possible prediction space of ligand-receptor interactions across 110 cell-type pairs, 28 samples, and 3 conditions. (2) Incorporation of spatially associated cell-type pairs (III). (3) Inclusion of information about spatially-colocalised ligand-receptor interactions (LRs), identified using (I) local scores with non-negative matrix factorisation (II). (4) Multi-view factor analysis using LRs predicted from single-cell data and informed by the spatial analysis. (5) Targeted differential expression analysis to find significantly deregulated LRs in a specific pair of interacting literature-supported cell types (‡). (6) A causal network search to generate a specific mechanistic hypothesis linking LR interactions to downstream Transcription Factors (TFs). The TF activities were estimated using a curated TF-target network¹¹ (IV) and linked to the LRs using a protein-protein interaction network¹². **B)** Mean silhouette scores of clustering heart failure versus healthy myeloid samples using genes extracted from CCC predictions across the distinct steps shown in subpanel A. Silhouette scores were z-transformed using randomly permuted gene sets of the same size as a baseline. The red dashed line signifies a z-score of 1.645, equivalent to a significant one-sided z-test. **C)** Adjusted Rand Index (ARI) across the distinct steps shown in subpanel A. The central line within each box marks the median, with the box hinges representing the first and third quartiles. The whiskers extend up to 1.5 times the interquartile range above and below the box hinges. In the evaluation, Steps 1 and I were respectively represented by prediction averages from CellPhoneDB P-values and SpatialDM's Global Moran's R across samples (Methods). **D)** Histogram of ARI values for Step 6, with one-sided empirical p-values calculated against a random distribution generated using sets of genes of the same size as the predictions in each step. The grey boxes depict the frequency of values drawn at random; the blue dashed line corresponds to the actual value.

Moreover, we apologise for not communicating clearly what we meant by “*it is not possible with current benchmark data to say which method performs best*”, and for possibly being over-cautious with interpreting the results of our evaluations - which are, to our knowledge, the most comprehensive set of benchmarks in any cell-cell-communication manuscript. In addition to the reliability evaluation above, using other four independent types of indirect ground truth (tissue pathology, cell type specificity, spatial co-localisation, and condition classification), we have compared the performances of the single-cell and spatial components and the individual methods in LIANA+ (see **Supp. Figures 3, 6, and 7** below).

We have now adjusted our text to better highlight the results and insights from our benchmarks. Specifically, we highlight that our benchmarks have shown that the methods unique to LIANA+ (e.g. spatially-weighted cosine similarity, rank aggregate, or intercellular multi-view factorization) exhibit comparable or improved performance than the respective state-of-the-art methods. We summarize these results below, highlighting in blue the added text:

1) Our results have shown that the proposed local metrics in LIANA+ (e.g. cosine similarity) perform on average better than e.g. SpatialDM (Moran’s R) or scHOT (masked Spearman) when predicting malignancy labels and cell type presence (**Supplementary Figure 3**).

We have clarified this also for our choice of cosine similarity as LIANA+’s default metric in **Supp. Note 1**:

“[...] In summary, all spatially-informed local scores in LIANA+ **performed well at predicting malignancy and cell type specificity, suggesting they preserved the biological signal captured by gene expression.** Yet, our results suggested that spatially-weighted products, Jaccard, and cosine **similarity** performed **on average** best in both the regression and classification tasks, albeit marginally better than other methods. **From these well-performing local metrics, we chose cosine similarity as LIANA+’s default, also used throughout the manuscript, since it’s easily interpretable (being bound between -1 and +1) and does not require the data to be binarised. [...]**”

Supplementary Figure S3. A) AUROC and B) weighted F1 when using local metrics to classify malignant spots in breast cancer data ¹³; **C) R² and D) RMSE** when using local metrics to predict cell type proportions in heart data ⁹. The line in the boxplots represents the median with hinges showing the first and third quartiles and the whiskers extend up to 1.5 times the interquartile range above and below the box hinges.

2) We have used our co-localization task (**Supp Note. 2**) which expands and improves previous benchmarks in the single-cell field ^{14–16}. We also used it to underline strategies to refine predictions into potentially more reliable insights from cross-technology data, enabled by LIANA+ being a standalone framework.

In the same task, we have shown that all individual single-cell methods in LIANA+, including LIANA's Rank aggregate, proposed in our previous manuscript ¹⁶, perform generally better than random (**Supp. Figure 6**). We have now additionally highlighted that magnitude-based scoring functions (whose names are labeled in red in the figures' X-axis), such as LIANA's magnitude rank, perform better than functions that focus on interaction specificity across cell type pairs:

“[...] Moreover, our benchmark highlighted that scoring functions that focus on the magnitude of interactions (e.g. LIANA's magnitude rank and ligand-receptor means or products), provide superior performance than functions that reflect interaction specificity across cell type pairs (**Supp. Figure S6C&D**). [...]”

Supplementary Figure S6. Comparison of ligand-receptor inference methods using the spatial colocalisation of cell types and ligand-receptors as assumed truth. Each method and its individual scoring functions are represented by a different box colour. Scoring functions that reflect the magnitude strength of an interaction are shown in dark red, and those that capture cell-type specificity in black. **A)** Quantifies the performance of each method's scoring functions using the Area Under the Receiver Operating Characteristic Curve (AUROC). **B)** Balanced Accuracy for each of the methods filtered according to their suggested false positive filtering thresholds. **C)** and **D)** measure normalised F1 scores following filtering for each score (*) and method, respectively. For each metric, a score of 0.5, denoted by the dashed red line, indicates random performance. Note that while we show 0 to 1 for normalised F1, unlike Balanced Accuracy and AUROC, it is not bound to 1 (Methods). Both NATMI and Connectome use expression products (Product*) as a measure of magnitude strength. The boxes were ordered according to the median performance across datasets (central line in cyan within each box). The box hinges represent the first and third quartiles, and the whiskers extend up to 1.5 times the interquartile range above and below the box hinges. Outliers are depicted as individual hollow points beyond the whiskers.

3) We have shown that multi-view factorization has on average better classification performance than the current state-of-the-art¹⁷ across individual methods and datasets. With the added benefit that it reports interactions per cell type pair (not a single vector of loadings across all cell types), and can be efficiently run on a standard laptop without GPU acceleration:

We have highlighted the advantages of multi-view factorisation in the relevant **Supp Note 3**:

“[...] Taken together, in addition to performing on average better at capturing condition-relevant variance, our proposed approach, using multi-view factorisation to extract intercellular programmes, uniquely provides interaction importances per cell-type pair and is highly-efficient, without the need for GPU acceleration (**Supp. Figure S1D**; **Supp. Table 1**).
[...]

Supplementary Figure S7. A) Classification setup to evaluate the ability of ligand-receptor methods, combined with Tensor-cell2cell and multi-view factor analysis (MOFA+) to separate conditions from multi-condition atlases in an unsupervised manner. **B)** Area under the receiver-operator curve (AUROC) and **C)** weighted F1 score, as calculated across 5 datasets, and the 'Average' performance.

2. The authors should better demonstrate LIANA+'s scalability by displaying the total running time and memory usage of an entire workflow of LIANA+, including individual methods running and Bayesian multi-view factor analysis.

We agree with the reviewer that it is important to better demonstrate the scalability of LIANA+. We have now included running time and memory usage for seven start-to-finish workflows (**Supp. Table 1**). All of these were executed on a personal laptop from 2019, equipped with 16GB RAM and an INTEL i7 processor. The maximum RAM usage did not exceed 4GB, with the longest workflow running time being under 10 minutes.

Of note, the application of multi-view factor analysis to CCC stands out for its computational efficiency (2 minutes on a standard laptop), outperforming the state-of-the-art¹⁷ which took 9 minutes to complete the same workflow with GPU acceleration. Similarly, the integration of spatial multi-omics data using multi-view learning and local bivariate metrics was completed in just 3 minutes.

Supplementary table 1. *Computational running time of standard LIANA+ workflows.*

Workflow	Data Details	Method Components	Total RAM usage†	Running Time†
Steady-state single-cell data analysis	3k PBMCs	LIANA single-cell Rank aggregate* of 5 methods with 1,000 permutations, visualization	< 1GB	< 1 minute
Identification of bivariate spatial co-expression patterns	A myocardial infarction 10X Visium slide	Estimation of global and local co-localisation statistics, corresponding categories and permutation p-values. Subsequent, decomposition of local scores into spatial CCC patterns using NMF, Pathway enrichment and co-localisation with TFs inferred to be active.	2.5GB	6 minutes
Learning spatial relationships across views	A myocardial infarction 10X Visium slide	Exploratory analysis leveraging linear and random forest models in diverse multi-view modeling designs.	2GB	4 minutes
Learning spatial relationships and identifying local bivariate associations from multi-omics data	Joint 10X Visium and MALDI-MSI data (~3k spots each)	Multi-view learning and bivariate spatial associations on spatial multi-omics data	1GB	3 minutes
Decomposition of CCC patterns with Bayesian Multi-view Factor Analysis	16 samples from Lupus patients, with a total of 15k cells	Rank aggregate* of 5 single-cell methods, multi-view factorisation, Pathway enrichment	1GB	2 minutes
Decomposition of CCC patterns with Tensor-cell2cell	16 samples from Lupus patients, with a total of 15k cells	Rank aggregate* of 5 single-cell methods, PARAFAC factorisation, Pathway enrichment	3.5GB	9 minutes (with GPU)
Differential expression analysis, with causal network inference	16 samples from Lupus patients, with a total of 15k cells	Identification of DE ligand-receptor interactions, inference of putative causal connections to downstream TFs	1.5GB	3.5 minutes
* LIANA's rank aggregate, published in our previous work ¹⁶ , can be replaced by any of the other 8 single-cell ligand-receptor methods in LIANA+, see Supp. Table 3 .				
† Running time and RAM statistics were generated using the start-to-finish workflows of LIANA+ (v1.0.5) on a personal				

laptop (Dell XPS15 2019), with an Intel Processor i7-9750H and 16GB of RAM. The workflows are also available as vignettes on LIANA+'s webpage: <https://liana-pv.readthedocs.io/>.

Since one of the novelty of LIANA+ is integrating multi-omics data, its scalability while applying to data with different numbers of views should also be evaluated.

We agree with the reviewer. We have now benchmarked the computational efficiency of multi-view factor analysis for CCC and our spatial multi-view learning approach across different numbers of views.

We evaluated multi-view learning in a typical task where we want to find the variables that best explain certain cell types, in this case in an ischemic heart sample. We saw that time-wise our multi-view learning approach scaled linearly with the number of views, with fitting 10 extra views taking ~ 3 minutes per run. It also had a relatively constant RAM usage of around ~2,9GB, regardless of the number of views (**Supp. Fig S1C**).

Moreover, since LIANA+'s last update, users can pass external single-view model algorithms, including those that work with GPUs.

While previous work has shown that multi-view factor analysis is fast and efficient¹⁸, we have now evaluated its efficiency with respect to the 'extreme' numbers of views in our case (typically not anticipated in multi-omics analyses). In our use case, views represent individual cell-type pairs, and we have shown that up to 400 views, multi-view factorisation takes < 8 minutes, with realistic scenarios (16-64 cell-type pairs), taking less than 2 minutes and < 3GB of RAM (**Supp. Fig S1D**).

Supplementary Figure S1. Efficiency benchmark of **A)** dissociated methods and **B)** spatially-weighted local metrics implemented in LIANA+. Time in *log₂ of seconds* and peak total RAM (memory) usage are shown over a number of simulated datasets with observations (cells) ranging from 1000 to 100,000 (1k to 100k). Masked Spearman Correlation was excluded for the 100k dataset. **C)** Multi-view learning of cell-type spatial patterns across a range of views. RAM Usage is not shown as it was relatively constant at ~2,900MB (± 100 MB) across views. **D)** Multi-view factor analysis used to identify intercellular programmes across a range of cell type pairs (views). The median time in minutes and RAM used are shown across five independent runs for each method.

Minor comments:

1. Add a significant level to Fig 2E.

We performed a paired two-sided t-test and added the significance level to the plot, which was moved to **Supp. Figure S4C** to address Comment #7 by Reviewer 1.

2. Scale the y-axis and adjust points to make Supplementary Figure S1 be more recognizable.

We have scaled the time in seconds and adjusted the y-axis such that the points are visible (see **Supp. Fig. S1** above).

3. The boxes in Supplementary Figure S6 should be ranked to be more informative.

We have ranked the boxes in **Supp. Fig. S6** according to their median performance (see response to Major Comment 1).

**Additional comments from reviewer#2 after assessing revision plan:

I have now had the opportunity to thoroughly review the authors' proposed revision plan.

Upon reviewing the attached file containing the authors' responses to my previous comments and concerns, I am pleased to note that the authors have presented a strategic approach for generating a reliable interaction set, which would allow for a comprehensive evaluation of LIANA+. If the proposed analyses are implemented as outlined by the authors, then my concerns about the integration benchmark can probably be addressed. Regarding scalability, the authors provided a detailed summary, and I am surprised by the high computational efficiency of multi-view factor analysis. If the approximate running time remains relatively unchanged during the formal run, then I am confident that my concerns in this area will also be adequately addressed.

In summary, I find the authors' proposal to be compelling and capable of addressing the issues I raised during my re-review. I believe these suggested changes would constitute a minor revision that significantly enhances the quality of the manuscript.

We thank the reviewer for the positive assessment of our revisions plan, and we have now carried out all requested analyses, which we think strengthen the manuscript. The newly-added benchmarks support the LIANA+'s ability to generate sets of reliable interactions and the efficiency of its multi-view components. As such, we hope that these additions have sufficiently underlined the value of our framework and manuscript; thereby addressing the remaining concerns of the reviewer.

Bibliography

1. Vicari, M. *et al.* Spatial multimodal analysis of transcriptomes and metabolomes in tissues. *Nat. Biotechnol.* (2023) doi:10.1038/s41587-023-01937-y.
2. Biancalani, T. *et al.* Deep learning and alignment of spatially resolved single-cell transcriptomes with Tangram. *Nat. Methods* **18**, 1352–1362 (2021).
3. Zeisel, A. *et al.* Molecular architecture of the mouse nervous system. *Cell* **174**, 999-1014.e22 (2018).
4. Farr, E. B. *et al.* MetalinksDB: a flexible and contextualizable resource of metabolite-protein interactions. *BioRxiv* (2023) doi:10.1101/2023.12.30.573715.
5. Kalia, L. V. & Lang, A. E. Parkinson's disease. *Lancet* **386**, 896–912 (2015).
6. Li, B. *et al.* Benchmarking spatial and single-cell transcriptomics integration methods for transcript distribution prediction and cell type deconvolution. *Nat. Methods* **19**, 662–670 (2022).
7. Andersson, A. *et al.* Single-cell and spatial transcriptomics enables probabilistic inference of cell type topography. *Commun. Biol.* **3**, 565 (2020).
8. Wang, W., Ren, X., Chen, X., Hong, Q. & Cai, G. Integrin β 1-rich extracellular vesicles of kidney recruit Fn1⁺ macrophages to aggravate ischemia-reperfusion-induced inflammation. *JCI Insight* (2024).
9. Kuppe, C. *et al.* Spatial multi-omic map of human myocardial infarction. *Nature* **608**, 766–777 (2022).
10. Hoefft, K. *et al.* Platelet-instructed SPP1⁺ macrophages drive myofibroblast activation in fibrosis in a CXCL4-dependent manner. *Cell Rep.* **42**, 112131 (2023).
11. Müller-Dott, S. *et al.* Expanding the coverage of regulons from high-confidence prior knowledge for accurate estimation of transcription factor activities. *Nucleic Acids Res.* **51**, 10934–10949 (2023).
12. Türei, D. *et al.* Integrated intra- and intercellular signaling knowledge for multicellular omics analysis. *Mol. Syst. Biol.* **17**, (2021).

13. Wu, S. Z. *et al.* A single-cell and spatially resolved atlas of human breast cancers. *Nature Genetics* (2021).
14. Liu, Z., Sun, D. & Wang, C. Evaluation of cell-cell interaction methods by integrating single-cell RNA sequencing data with spatial information. *Genome Biol.* **23**, 218 (2022).
15. Luo, J., Deng, M., Zhang, X. & Sun, X. ESICCC as a systematic computational framework for evaluation, selection, and integration of cell-cell communication inference methods. *Genome Res.* (2023) doi:10.1101/gr.278001.123.
16. Dimitrov, D. *et al.* Comparison of methods and resources for cell-cell communication inference from single-cell RNA-Seq data. *Nat. Commun.* **13**, 3224 (2022).
17. Armingol, E. *et al.* Context-aware deconvolution of cell-cell communication with Tensor-cell2cell. *Nat. Commun.* **13**, 3665 (2022).
18. Argelaguet, R. *et al.* MOFA+: a statistical framework for comprehensive integration of multi-modal single-cell data. *Genome Biol.* **21**, 111 (2020).

Decision Letter, first revision:

Our ref: NCB-A53738A

3rd June 2024

Dear Dr. Saez Rodriguez,

Thank you for submitting your revised manuscript "LIANA+: an all-in-one cell-cell communication framework" (NCB-A53738A). It has now been seen by the original Referee #2 (who kindly assessed your responses to Rev#1's comments as well) and their comments are below. The reviewer finds that the paper has improved in revision, and therefore we'll be happy in principle to publish it in Nature Cell Biology, pending minor revisions to comply with our editorial and formatting guidelines.

Please note: the current version of your manuscript is in a PDF format; could you please email us a copy of the file in an editable format (Microsoft Word or LaTeX), as we cannot proceed with PDFs at this stage? Many thanks for your attention to this point.

With the Word file in-hand, we will be performing detailed checks on your paper and will send you a checklist detailing our editorial and formatting requirements in about 2 weeks. Please do not upload the final materials and make any revisions until you receive this additional information from us.

Thank you again for your interest in Nature Cell Biology. Please do not hesitate to contact me if you have any questions.

Sincerely,

Melina

Melina Casadio, PhD
Senior Editor, Nature Cell Biology
ORCID ID: <https://orcid.org/0000-0003-2389-2243>

Reviewer #2 (Remarks to the Author):

The authors performed a further benchmark on the contribution of integration strategy in LIANA+ where the results show that the combination of individual methods indeed improved the performance. Moreover, the multi-view factor analysis also displayed high computational efficiency in the formal run. In the end, the strengthened manuscript addressed all my remaining concerns.

Decision Letter, final checks:

Our ref: NCB-A53738A

14th June 2024

Dear Dr. Saez Rodriguez,

Thank you for your patience as we've prepared the guidelines for final submission of your Nature Cell Biology manuscript, "LIANA+: an all-in-one cell-cell communication framework" (NCB-A53738A). Please carefully follow the step-by-step instructions provided in the attached file, and add a response in each row of the table to indicate the changes that you have made. Ensuring that each point is addressed will help to ensure that your revised manuscript can be swiftly handed over to our production team.

In recognition of the time and expertise our reviewers provide to Nature Cell Biology's editorial process, we would like to formally acknowledge their contribution to the external peer review of your manuscript entitled "LIANA+: an all-in-one cell-cell communication framework". For those reviewers who give their assent, we will be publishing their names alongside the published article.

Nature Cell Biology offers a Transparent Peer Review option for new original research manuscripts submitted after December 1st, 2019. As part of this initiative, we encourage our authors to support increased transparency into the peer review process by agreeing to have the reviewer comments, author rebuttal letters, and editorial decision letters published as a Supplementary item. When you submit your final files please clearly state in your cover letter whether or not you would like to participate in this initiative. Please note that failure to state your preference will result in delays in accepting your manuscript for publication.

Cover suggestions

COVER ARTWORK: We welcome submissions of artwork for consideration for our cover. For more information, please see our guide for cover artwork.

Nature Cell Biology has now transitioned to a unified Rights Collection system which will allow our Author Services team to quickly and easily collect the rights and permissions required to publish your work. Approximately 10 days after your paper is formally accepted, you will receive an email in providing you with a link to complete the grant of rights. If your paper is eligible for Open Access, our Author Services team will also be in touch regarding any additional information that may be required

to arrange payment for your article.

Please note that *Nature Cell Biology* is a Transformative Journal (TJ). Authors may publish their research with us through the traditional subscription access route or make their paper immediately open access through payment of an article-processing charge (APC). Authors will not be required to make a final decision about access to their article until it has been accepted. Find out more about Transformative Journals

Please use the following link for uploading these materials:
[Redacted]

Best regards,

Aimee Frier
Staff
Nature Cell Biology

On behalf of

Melina Casadio, PhD
Senior Editor, Nature Cell Biology
ORCID ID: <https://orcid.org/0000-0003-2389-2243>

Reviewer #2:

Remarks to the Author:

The authors performed a further benchmark on the contribution of integration strategy in LIANA+

where the results show that the combination of individual methods indeed improved the performance. Moreover, the multi-view factor analysis also displayed high computational efficiency in the formal run. In the end, the strengthened manuscript addressed all my remaining concerns.

Author Rebuttal, first revision:

Reviewer #2:

Remarks to the Author:

The authors performed a further benchmark on the contribution of integration strategy in LIANA+ where the results show that the combination of individual methods indeed improved the performance. Moreover, the multi-view factor analysis also displayed high computational efficiency in the formal run. In the end, the strengthened manuscript addressed all my remaining concerns.

We thank the reviewer for the positive review and for their critical comments that helped us improve our work.

Final Decision Letter:

Dear Dr Saez Rodriguez,

I am pleased to inform you that your manuscript, "LIANA+ provides an all-in-one framework for cell-cell communication inference", has now been accepted for publication in Nature Cell Biology.

Please note that *Nature Cell Biology* is a Transformative Journal (TJ). Authors may publish their research with us through the traditional subscription access route or make their paper immediately open access through payment of an article-processing charge (APC). Authors will not be required to make a final decision about access to their article until it has been accepted. Find out more about Transformative Journals

If your paper includes color figures, please be aware that in order to help cover some of the additional cost of four-color reproduction, Nature Portfolio charges our authors a fee for the printing of their color

figures. Please contact our offices for exact pricing and details.

If you have not already done so, we strongly recommend that you upload the step-by-step protocols used in this manuscript to protocols.io (<https://protocols.io>), an open online resource that allows researchers to share their detailed experimental know-how. All uploaded protocols are made freely available and are assigned DOIs for ease of citation. Protocols and Nature Portfolio journal papers in which they are used can be linked to one another, and this link is clearly and prominently visible in the online versions of both. Authors who performed the specific experiments can act as primary authors for the Protocol as they will be best placed to share the methodology details, but the Corresponding Author of the present research paper should be included as one of the authors. By uploading your Protocols onto protocols.io, you are enabling researchers to more readily reproduce or adapt the methodology you use, as well as increasing the visibility of your protocols and papers. You can also establish a dedicated workspace to collect your lab Protocols. Further information can be found at <https://www.protocols.io/help/publish-articles>.

With kind regards,

Melina Casadio, PhD
Senior Editor, Nature Cell Biology
ORCID ID: <https://orcid.org/0000-0003-2389-2243>

** Visit the Springer Nature Editorial and Publishing website at www.springernature.com/editorial-and-publishing-jobs for more information about our career opportunities. If you have any questions please click here.**